# PISA Experiments: Exploring Physics Post-Training for Video Diffusion Models by Watching Stuff Drop

**Chenyu Li** [* 1]  **Oscar Michel** [* 1]  **Xichen Pan** [1]  **Sainan Liu** [2]  **Mike Roberts** [2]  **Saining Xie** [1]

## Abstract

Large-scale pre-trained video generation models excel in content creation but are not reliable as physically accurate world simulators out of the box. This work studies the process of post-training these models for accurate world modeling through the lens of the simple, yet fundamental, physics task of modeling object freefall. We show state-of-the-art video generation models struggle with this basic task, despite their visually impressive outputs. To remedy this problem, we find that fine-tuning on a relatively small amount of simulated videos is effective in inducing the dropping behavior in the model, and we can further improve results through a novel reward modeling procedure we introduce. Our study also reveals key limitations of post-training in generalization and distribution modeling. Additionally, we release a benchmark for this task that may serve as a useful diagnostic tool for tracking physical accuracy in large-scale video generative model development. Code is available at this repository: https://github.com/vision-x-nyu/pisa-experiments.

## 1. Introduction

Over the past year, video generation models have advanced significantly, inspiring visions of a future where these models could serve as realistic world models (Craik, 1967; LeCun, 2022; Hafner et al., 2019; 2023; Ha & Schmidhuber, 2018). State-of-the-art video generation models models exhibit impressive results in content creation (OpenAI, 2024; Kuaishou, 2024; Luma, 2024; Runway, 2024) and are already being used in advertising and filmmaking (Runway, 2025; NBC, 2025). These advancements have sparked a line of research that seeks to evolve these models from content creators to world simulators for embodied agents (Yang et al., 2023; 2024b; Agarwal et al., 2025). However, accurate world modeling is considerably more challenging than creative content creation because looking "good enough" is not sufficient: generated pixels must faithfully represent a world state evolving in accordance with the laws of physics and visual perspective.

We find that although the generations of state-of-the-art models are impressive *visually*, these models still struggle to generate results that are accurate *physically*, even though these models are pretrained on internet-scale video data demonstrating a wide variety of complex physical interactions. The failure to ground and align visual generations to the laws of physics suggests that pretraining is not enough and a post-training stage is needed. Much like how pretrained Large Language Models (LLMs) need to be adapted through post-training before they can be useful conversational assistants, pretrained video generative models ought to be adapted through post-training before they can be deployed as physically accurate world simulators.

In this work, we rigorously examine the post-training process of video generation models by focusing on the simple yet fundamental physics task of **modeling object freefall**, which we find is highly challenging for state-of-the-art models. Specifically, we study an image-to-video[1] (I2V) scenario where the goal is to generate a video of an object falling and potentially colliding with other objects on the ground, starting from an initial image of the object suspended in midair. We chose to study this *single task*, rather than general physics ability as a whole, because its simplicity allows us to conduct controlled experiments that yield insights into the strengths and limitations of the post-training process, which we believe will become an increasingly important component of research in generative world modeling. Additionally, the simplicity of the dropping task allows it to be implemented in simulation which is desirable because it allows us to easily test the properties of dataset scaling, gives us access to ground truth annotations for evaluation, and gives us the ability to precisely manipulate the simulation

---

[*]Equal contribution, alphabetical order. [1]New York University [2]Intel Labs. Correspondence to: Saining Xie <saining.xie@nyu.edu>.

*Proceedings of the 42^{nd} International Conference on Machine Learning*, Vancouver, Canada. PMLR 267, 2025. Copyright 2025 by the author(s).

---

[1]We discuss our decision to formulate this task in the image-to-video setting instead of the video-to-video setting in Appendix A.

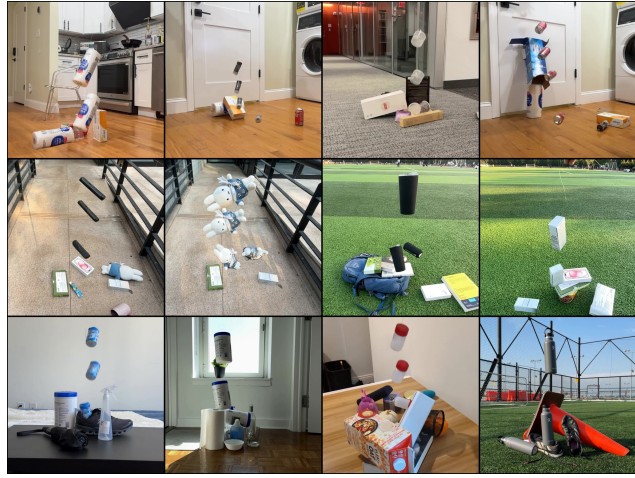 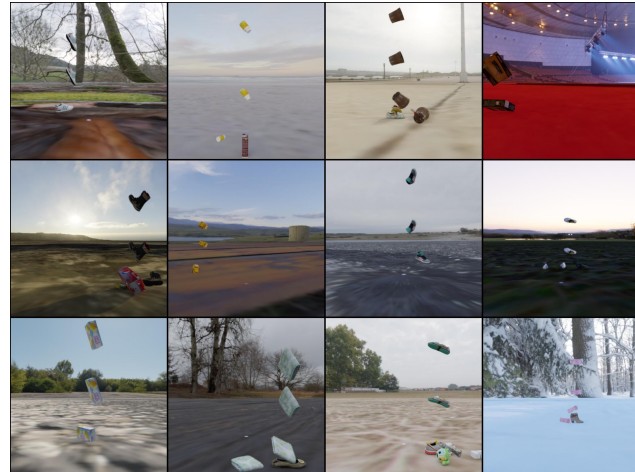

*Figure 1.* Our PISA (Physics-Informed Simulation and Alignment) evaluation framework includes a new video dataset, where objects are dropped in a variety of real-world (**left**) and synthetic (**right**) scenes. For visualization purposes, we depict object motion by overlaying multiple video frames in each image shown above. Our real-world videos enable us to evaluate the physical accuracy of generated video output, and our synthetic videos enable us to improve accuracy through the use of post-training alignment methods.

environment for *controlled experimentation*.

Named after Galileo's famous dropping experiment, we introduce the PISA (Physics-Informed Simulation and Alignment) framework for studying physics post-training in the context of the dropping task. PISA includes new real and simulated video datasets, as shown in Figure 1, containing a diverse set of dropping scenarios. PISA also includes a set of task-specific metrics that focus on measuring physical accuracy. Our real-world videos and metrics enable us to evaluate the physical accuracy of generated video output, and our synthetic videos enable us to improve accuracy through a post-training process we introduce.

Our study reveals that current state-of-the-art video generative models struggle significantly with the task of physically accurate object dropping. Generated objects frequently exhibit impossible behaviors, such as floating midair, defying gravity, or failing to preserve realistic trajectories during freefall. However, we find that simple fine-tuning can be remarkably effective: fine-tuning an open-source model on a small dataset of just a few thousand samples enables it to vastly outperform state-of-the-art models in physical accuracy. We further observe that pretrained models are critical for success; models initialized randomly, without leveraging pretraining on large-scale video datasets, fail to achieve comparable results. We also introduce a novel framework for reward modeling that yields further improvement. We demonstrate that our reward learning system is highly flexible in that different reward functions can be chosen to target different axes of physical improvement.

Our analysis also reveals key limitations. First, we see that model performance degrades when tasked with scenarios outside the training distribution, such as objects dropping from unseen depths or heights. Additionally, while our post-trained model generates object motion that is 3D-consistent and physically accurate, we observe misalignment between the generated and ground truth dropping time distribution.

These findings indicate that post-training is likely to be an essential component of future world modeling systems. The challenges we identify in this relatively simple task are likely to persist when modeling more sophisticated physical phenomena. By introducing the PISA framework and benchmark, we provide a useful diagnostic tool for researchers to test whether models are on the path to acquiring general physical abilities, as well as identify key limitations that researchers should be aware of when integrating new capabilities into their models through post-training.

## 2. Related Work

**Modeling Intuitive Physics.** Intuitive physics refers to the innate or learned human capacity to make quick and accurate judgments about the physical properties and behaviors of objects in the world, such as their motion, stability, or interactions. This ability, present even in infancy (Spelke et al., 1992; Baillargeon, 2004; Battaglia et al., 2013), is crucial for navigating and understanding everyday life. Replicating intuitive physics is a foundational step toward creating systems that can interact effectively and safely in dynamic, real-world environments (Lake et al., 2017). Gravity, as a core component of intuitive physics, plays a pivotal role in both domains. It is one of the most universal and observable physical forces, shaping our expectations about object motion, stability, and interaction (Hamrick et al., 2016; Ullman

et al., 2017). Many studies in cognitive science (Battaglia et al., 2013) and AI (Wu et al., 2015; Bear et al., 2021) have relied on physics engines to evaluate and model intuitive physics. Our work uses the Kubric engine (Greff et al., 2022) to generate training videos.

**Video Generation Models as World Simulators.** Video generation has long been an intriguing topic in computer vision, particularly in the context of predicting future frames (Srivastava et al., 2015; Xue et al., 2016). More recently, as large-scale generative models have become prominent, Yang et al. explored how a wide range of real-world dynamics and decision-making processes can be expressed in terms of video modeling (Yang et al., 2024b; 2023). The introduction of the Sora model (OpenAI, 2024) marked a leap in the quality of generated videos and ignited interest in leveraging such models as "world simulators." Over the past year, numerous video generation models have emerged, some open-source (Zheng et al., 2024; Yang et al., 2024c; Jin et al., 2024; Agarwal et al., 2025) and others commercially available (Kuaishou, 2024; Luma, 2024; Runway, 2024; OpenAI, 2024). Related to our work, Kang et al. (Kang et al., 2024) study the extent to which video generation models learn generalizable laws of physics when trained on 2D data from a synthetic environment.

**Evaluating Video Generation Models.** Traditional image-based metrics for generative modeling, such Fréchet inception distance (FID) (Heusel et al., 2017) or inception score (IS) (Salimans et al., 2016), can be incorporated into the video domain, either by applying them on a frame-by-frame basis or by developing video-specific versions, such as Fréchet video distance (FVD) (Unterthiner et al., 2018). Going beyond distribution matching measures, several benchmarks have developed suites of metrics that aim to better evaluate the semantic or visual quality of generated videos. For example, V-Bench (Huang et al., 2024) offers a more granular evaluation by measuring video quality across multiple dimensions, such as with respect to subject consistency or spatial relationships. In physics, some works, such as VideoPhy (Bansal et al., 2024) and PhyGenBench (Meng et al., 2024), evaluate in the T2V setting by utilizing multimodal large language models (MLLM) to generate a VQA-based score. More recently, Cosmos (Agarwal et al., 2025) and Physics-IQ (Motamed et al., 2025), evaluate physics in the image-to-video and video-to-video settings.

## 3. PisaBench

Our benchmark, *PisaBench*, examines the ability of video generative models to produce accurate physical phenomena by focusing on a straightforward dropping task.

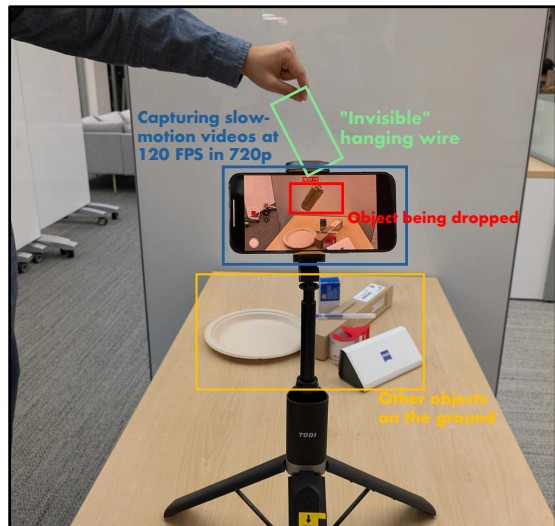

*Figure 2.* The setup for collecting real-world videos.

### 3.1. Task Definition & Assumptions

Our task can be summarized as follows: given an image of an object suspended in midair, generate a video of the object falling and colliding with the ground and potentially other objects. Since a video is an incomplete partial observation of the 4D world, we make a number of assumptions to constrain the task space. These assumptions are crucial for ensuring that our metrics are reliable signals for physical accuracy, since they are only approximations of task success computed from a single ground truth and generated video.

Specifically, we assume that the falling object is completely still in the initial frame, that only the force of gravity is acting on the object while it falls, and that the camera does not move. The first two assumptions are necessary for the image-to-video setting. Since we do not provide multiple frames as input, it is otherwise impossible to establish the initial velocity or acceleration of the falling object without these assumptions. The last assumption is necessary as our metrics derive from the motion of segmentation masks, which would be affected in the presence of camera motion.

### 3.2. Real World Data

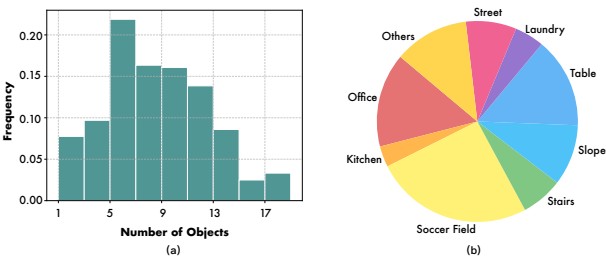

*Figure 3.* Statistics of the real-world data: (a) number of objects in each video, (b) the proportions of different scenes in the videos.

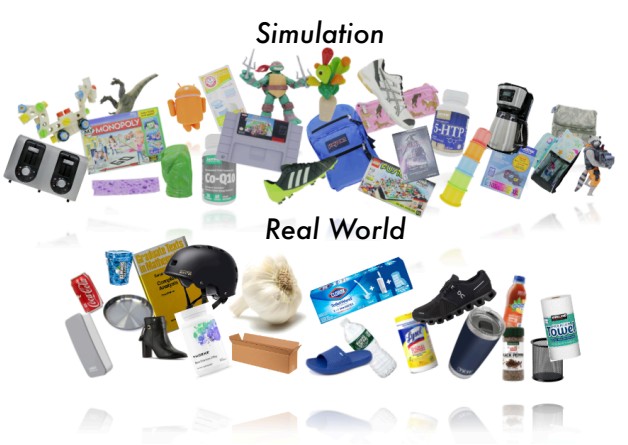

*Figure 4.* Examples of various objects included in our dataset. For simulation, we utilize the GSO dataset (Downs et al., 2022), while for the real-world dataset, we curate our own set of common household objects.

**Real World Videos.** We collect a set of 361 real-world videos demonstrating the dropping task for evaluation. As is shown in Figure 4, the dataset includes a diverse set of objects with different shapes and sizes, captured across various settings such as offices, kitchens, parks, and more (see Figure 3). Each video begins with an object suspended by an invisible wire in the first frame, which is necessary to enforce the assumption that objects are stationary at the start of the video. This assumption is required in our image-to-video setting; otherwise, the initial velocity of an object is ambiguous. We cut the video clips to begin as soon as the wire is released. We record the videos in slow-motion at 120 frames per second (fps) with cellphone cameras mounted on tripods to eliminate camera motion. An example of our video collection setup is show in Figure 2. Additional details on our collection system are provided in Appendix H.

**Simulated Test Videos.** Since our post-training process uses a dataset of simulated videos, we also create a simulation test-set of 60 videos for understanding sim2real transfer. We create two splits of 30 videos each: one featuring objects and backgrounds seen during training, and the other featuring unseen objects and backgrounds. See Section 4.1 for details on how our simulated data is created.

**Annotations.** As is shown in Figure 5, we annotate each video with a caption and segmentation masks estimated from the SAM 2 (Ravi et al., 2024) video segmentation model. We create a descriptive caption for each object in the format of "{object description} falls." This caption is used to provide context to the task when text input is supported.

### 3.3. Metrics

We propose three metrics to assess the accuracy of trajectories, shape fidelity, and object permanence. Each of our met-

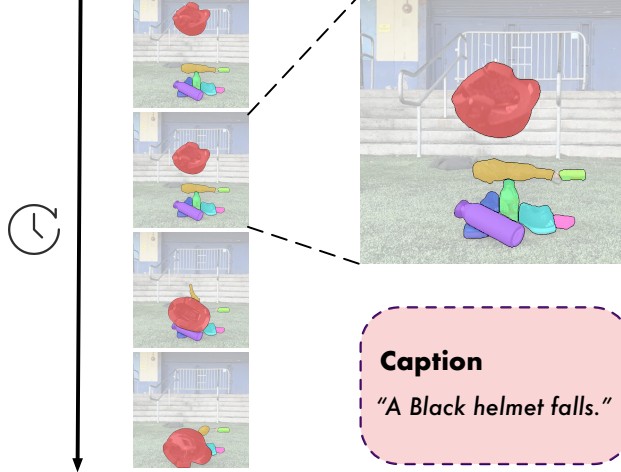

*Figure 5.* Example of annotations in real-world data. For segmentation masks, we manually annotate first frame and utilize SAM 2 to produce segmentation masks across frames. For captions, we annotate "{object description} falls." for all video segments.

rics compare frames from the ground-truth video with the generated video. Further details about the metrics, including their formulas and our resampling procedure for accounting for differences in fps, is described in Appendix B.

**Trajectory L2.** For each frame in both the generated video and ground truth, we calculate the centroid of the masked region. After doing this, we compute the average $L_2$ distance between the centroids of corresponding frames.

**Chamfer Distance (CD).** To assess the shape fidelity of objects, we calculate the Chamfer Distance (CD) between the mask regions of the generated video and ground truth.

**Intersection over Union (IoU).** We use the Intersection over Union (IoU) metric to evaluate object permanence. The IoU measures objects' degree of overlap between the generated video and ground truth.

### 3.4. Evaluation Results

We evaluate 4 open models including CogVideoX-5B-I2V(Yang et al., 2024c), DynamiCrafter(Xing et al., 2023), Pyramid-Flow(Jin et al., 2024), and Open-Sora-V1.2(Zheng et al., 2024), as well as 4 proprietary models including Sora (OpenAI, 2024), Kling-V1(Kuaishou, 2024), Kling-V1.5(Kuaishou, 2024), and Runway Gen3 (Runway, 2024). We also evaluate OpenSora post-trained through the processes of Supervised Fine-Tuning (PSFT) and Object Reward Optimization (ORO); see Section 4 for details.

The results of running the baseline models on the benchmark indicate a consistent failure to generate physically accurate dropping behavior, despite the visual realism of their generated frames. Qualitatively, we see common failure cases in Figure 6, such as implausible object deformations, float-

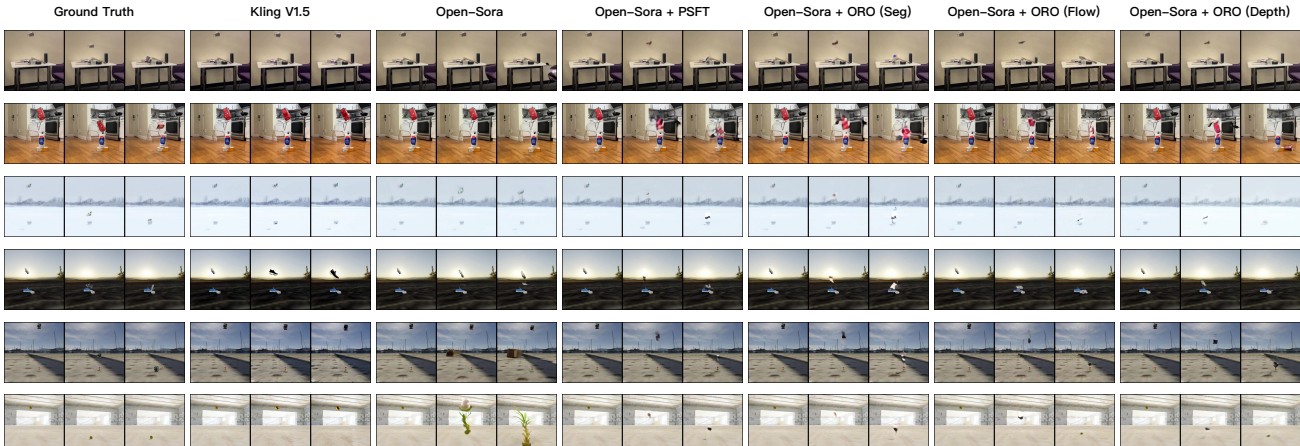

*Figure 6.* Qualitative comparison of results on real test set (row 1-2), simulated seen test set (row 3-4) and simulated unseen test set (row 5-6). We present the results of popular open-source and commercially available models alongside those of models fine-tuned through our method. Existing models often struggle to generate videos depicting objects falling, whereas our PSFT method effectively introduces knowledge of free-fall into the model. ORO enables the model to more accurately learn object motion and shape.

ing, hallucination of new objects, and unrealistic special effects. We further visualize a random subset of generated trajectories on the left of Figure 8. In many cases, the object remains completely static, and sometimes the object even moves upward. When downward motion is present, it is often slow or contains unrealistic horizontal movement.

## 4. Physics Post-Training

We present a post-training process to address the limitations of current models described in Section 3.4. We utilize simulated videos that demonstrate realistic dropping behavior. Our approach for post-training is inspired by the two-stage pipeline consisting of supervised fine-tuning followed by reward modeling commonly used in LLMs. We find that our pipeline improves performance on both real and simulated evaluations, with greater gains observed in simulation. This is due to the sim-to-real gap, though our approach still shows substantial gains in transferring to real-world data.

### 4.1. Simulated Adaptation Data

The first stage of our approach involves supervised fine-tuning. We use Kubric (Greff et al., 2022), a simulation and rendering engine designed for scalable video generation, to create simulated videos of objects dropping and colliding with other objects on the ground. Each video consists of 1-6 dropping objects onto a (possibly empty) pile of up to 4 objects underneath them. The videos are 2 seconds long, consisting of 32 frames at 16 fps. The objects are sourced from the Google Scanned Objects (GSO) dataset (Downs et al., 2022), which provides true-to-scale 3D models created from real-world scans across diverse categories (examples shown in Figure 4). The camera remains stationary in each video and is oriented parallel to the ground plane. To in-

troduce variability, we randomly sample the camera height between 0.4 and 0.6 meters and position objects between 1 and 3 meters away from the camera, which corresponds to the distributions observed in the real-world dataset. More information about the dataset can be found in Appendix K.

### 4.2. Physics Supervised Fine-Tuning (PSFT).

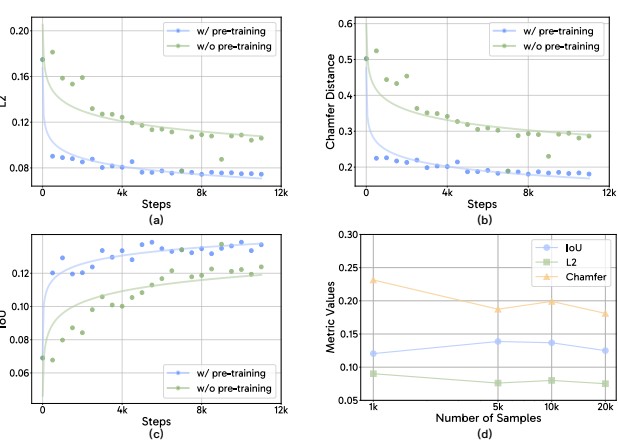

*Figure 7.* Plots (a), (b), and (c) demonstrate that our metrics tend to improve with further training and that leveraging a pre-trained video diffusion model enhances performance compared to random initialization. In plot (d), the size of the training dataset varies in each training run (each consisting of 5k steps). With only 5k samples, we can achieve optimal results.

We use the pretrained Open-Sora v1.2 (Zheng et al., 2024) model as our base model and fine-tune it on our simulated video dataset. We employ Open-Sora v1.2's rectified flow training objective without modification (Liu et al., 2022). Each fine-tuning experiment is conducted with a batch size of 128 and a learning rate of $1e{-}4$ on two 80GB NVIDIA

| | Method | Real | | | Sim (Seen) | | | Sim (Unseen) | | |
|---|---|---|---|---|---|---|---|---|---|---|
| | | L2 ($\downarrow$) | CD ($\downarrow$) | IoU ($\uparrow$) | L2 ($\downarrow$) | CD ($\downarrow$) | IoU ($\uparrow$) | L2 ($\downarrow$) | CD ($\downarrow$) | IoU ($\uparrow$) |
| Proprietary | Sora (OpenAI, 2024) | 0.174 | 0.488 | 0.065 | 0.149 | 0.446 | 0.040 | 0.140 | 0.419 | 0.031 |
| | Kling-V1 (Kuaishou, 2024) | 0.157 | 0.425 | 0.056 | 0.142 | 0.415 | 0.032 | 0.145 | 0.437 | 0.028 |
| | Kling-V1.5 (Kuaishou, 2024) | 0.155 | 0.424 | 0.058 | 0.137 | 0.396 | 0.033 | 0.132 | 0.405 | 0.029 |
| | Runway Gen3 (Runway, 2024) | 0.187 | 0.526 | 0.042 | 0.170 | 0.509 | 0.040 | 0.149 | 0.460 | 0.038 |
| Open | CogVideoX-5B-I2V (Yang et al., 2024c) | 0.138 | 0.366 | 0.080 | 0.112 | 0.315 | 0.020 | 0.101 | 0.290 | 0.020 |
| | DynamiCrafter (Xing et al., 2023) | 0.187 | 0.504 | 0.021 | 0.157 | 0.485 | 0.039 | 0.136 | 0.430 | 0.033 |
| | Pyramid-Flow (Jin et al., 2024) | 0.175 | 0.485 | 0.062 | 0.126 | 0.352 | 0.059 | 0.130 | 0.381 | 0.048 |
| | Open-Sora (Zheng et al., 2024) | 0.175 | 0.502 | 0.069 | 0.139 | 0.409 | 0.036 | 0.130 | 0.368 | 0.034 |
| Ours | Open-Sora + PSFT (**base**) | 0.076 | 0.188 | 0.139 | 0.036 | 0.088 | 0.165 | 0.028 | 0.058 | 0.129 |
| | **base** + ORO (Seg) | 0.075 | 0.183 | 0.142 | 0.033 | 0.076 | 0.170 | 0.032 | 0.063 | 0.145 |
| | **base** + ORO (Flow) | 0.067 | 0.164 | 0.136 | 0.026 | 0.062 | 0.122 | 0.022 | 0.045 | 0.071 |
| | **base** + ORO (Depth) | 0.067 | 0.159 | 0.129 | 0.031 | 0.072 | 0.124 | 0.022 | 0.046 | 0.096 |

*Table 1.* PisaBench Evaluation Results. This table compares the performance of four proprietary models, four open models, and the models fine-tuned with PSFT and PSFT + ORO on our real-world and simulated test set which is decomposed into seen and unseen object splits. Across all metrics, our PSFT models outperform all other baselines, including proprietary models like Sora. Reward modeling further enhances results, with segmentation rewards improving the shape-based IoU metric and optical rewards and depth rewards enhancing the motion-based L2 and CD metrics. This suggests that rewards can be flexibly adjusted to target specific aspects of performance.

A100 GPUs. As shown in Figure 6, fine-tuning with this data alone is sufficient to induce realistic dropping behavior in the model. Quantitatively, our PSFT model substantially improves on both our simulated and real-world benchmark, as shown in Table 1. **Dataset size.** We conduct an ablation study on the number of training samples to understand the amount of data required for optimal performance on our benchmark. We create random subsets from 500 to 20,000 samples and train our model for 5,000 gradient steps on each subset. Notably, as shown in Figure 7, only 5,000 samples are needed to achieve optimal results. **Effect of pre-training.** Additionally, we investigate the impact of Open-Sora's pre-training on adaptation. We randomly initialize the Open-Sora's denoising network while keeping the pre-trained initialization of the compressor network and train the model on a dataset of 5k training samples. As shown in Figure 8, the learned knowledge from Open-Sora's pre-training plays a critical role in our task.

*Overall, using PSFT on only 5k samples is sufficient to push Open-Sora's performance past all other evaluated models, including state-of-the-art commercial video generators, by a wide margin. This is made possible by leveraging the knowledge from the sufficiently pre-trained base model.*

### 4.3. Object Reward Optimization (ORO)

In the second stage, we propose *Object Reward Optimization* (ORO) to use reward gradients to guide the video generation model toward generating videos where the object's

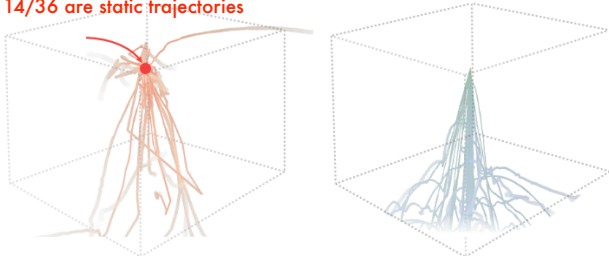

14/36 are static trajectories

*Figure 8.* On the left, we plot random trajectories from the baseline models in Table 1. On the right, we show random trajectories from our fine-tuned model. The baseline trajectories exhibit unrealistic behavior, and most of them stay completely static. On the right, we see the trajectories consistently falling downward with collision and rolling behavior being modeled after the point of contact.

motion and shape more closely align with the ground truth. We follow the VADER framework from (Prabhudesai et al., 2024) and introduce three reward models. The differences between our approach and VADER include: (1) our reward model utilizes both generated videos and ground truth instead of generated videos and conditioning. (2) gradients propagate through all denoising time steps in fine-tuning. Consequently, the VADER objective is modified as follows:

$$J(\theta) = \mathbb{E}_{(x_0,c)\sim\mathcal{D}, x_0'\sim p_\theta(x_0'|c)}[R(x_0', x_0)] \qquad (1)$$

where $\mathcal{D}$ is the ground truth dataset, $p_\theta(.)$ is a given video diffusion model, $x_0', x_0 \in \mathbb{R}^{H\times W\times 3}$ are generated video and ground truth, and $c \in \mathbb{R}^{H\times W\times 3}$ is the initial image.

**Segmentation Reward.** We utilize SAM 2 (Ravi et al., 2024) to generate segmentation masks across frames for generated videos. We define segmentation reward as the IoU between the dropping object's mask in generated video and the mask from the groud truth simulated segmentation.

**Optical Flow Reward.** We utilize RAFT (Teed & Deng, 2020) to generate generated video's optical flow $V^{\text{gen}}$ and ground truth's optical flow $V^{\text{gt}}$. We define the optical flow reward as $R(x_0', x_0) = -|V^{\text{gen}} - V^{\text{gt}}|$.

**Depth Reward.** We utilize Depth-Anything-V2 (Yang et al., 2024a) to generate generated video's depth map $D^{\text{gen}}$ and ground truth's depth map $D^{\text{gt}}$. We define the optical flow reward as $R(x_0', x_0) = -|D^{\text{gen}} - D^{\text{gt}}|$.

Details on implementation can be found in Appendix C.

We begin from the checkpoint of the first stage, which is trained on 5,000 samples trained over 5,000 gradient steps. We then fine-tune the model with ORO on the simulated dataset, using a batch size of 1 and two 80GB NVIDIA A100 GPUs for each fine-tuning experiment. We set a learning rate of 1e−6 for segmentation reward and depth reward and 1e−5 for optical flow.

*As shown in Table 1, incorporating ORO in reward modeling further improves performance. Additionally, each reward function enhances the aspect of physicality that aligns with its intended purpose—segmentation rewards improve shape accuracy, while flow rewards and depth rewards improve motion accuracy. This demonstrates the process is both modular and interpretable.*

## 5. Assessing Learned Physical Behavior

Having introduced our post-training approaches in Section 4, we probe into the model's understanding of the interaction between gravity and perspective—the two laws that determine the dynamics of our videos. We first test if the learned physical behavior of our model can generalize to dropping heights and depths beyond its training distribution. Then, we study the ability of the model to learn the probability distribution induced by the uncertainty of perspective.

### 5.1. Generalization to Unseen Depths and Heights

Depth and height are the main factors that affect the dynamics of a falling object in our videos. We can see this by combining the laws of gravity with perspective under our camera assumptions to model the object's image $y$ coordinate as a function of time (further details on our coordinate system are described in Appendix D):

$$y(t) = \frac{f}{Z}\left(Y_0 - \frac{1}{2}gt^2\right). \tag{2}$$

From Equation (2), we see that the random variables that affect object motion are $Z$ (depth) and $Y$ (height) (the camera focal length $f$ is fixed). Thus, we are interested in testing generalization on unseen values of $Y$ and $Z$.

We create a simulated test set in which a single object is dropped from varying depths and heights, using objects and backgrounds unseen during training. We uniformly sample depth and height values (in meters) from the Cartesian product of the ranges $[1, 5]$ and $[0.5, 2.5]$, respectively. The camera height is fixed at $0.5m$, and depth-height pairs outside the camera viewing frustum are discarded. A sample is in-distribution (ID) if its dropping depth and height both fall in the range $[1, 3]$ and $[0.5, 1.5]$.

Since we have access to the ground truth dropping time in simulation, we also employ a dropping time error, a metric we describe in Appendix B. Our analysis in Table 2 shows that performance degrades for out-of-distribution scenarios.

*Since depth and height are the main physical quantities that affect falling dynamics, this finding indicates that our model may struggle to learn a fully generalizable law that accounts for the interaction of perspective and gravity.*

| Setting | L2 ($\downarrow$) | Chamfer ($\downarrow$) | IOU ($\uparrow$) | Time Error ($\downarrow$) |
|---|---|---|---|---|
| ID | 0.036 | 0.088 | 0.155 | 0.091 |
| OOD | 0.044 | 0.143 | 0.049 | 0.187 |

*Table 2.* Results of our metrics on in-distribution (ID) and out-of-distribution (OOD) depth-height combinations. The values used for depth range from 1-5m (ID range $[1, 3]$) and height values range from 0.5-2.5 (ID range $[0.5, 1.5]$).

### 5.2. Distributional Analysis

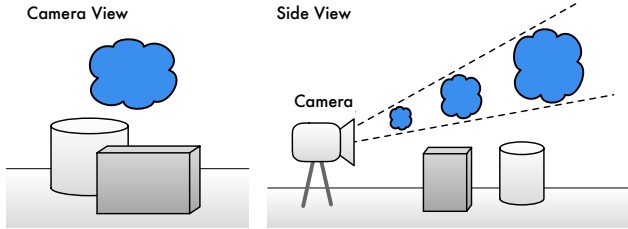

*Figure 9.* Demonstration of ambiguity in 2D perspective projections. Each of the three clouds appears the exact same in the camera's image. The right side shows how we perform a scale and translation augmentation to generate deliberately ambiguous data.

The evolution of a physical system is not uniquely determined by a single initial image, since the lossy uncertainty of perspective induces a distribution of possible outcomes as shown in Figure 9. An ideal video world model should (1) output videos that are faithful to the evolution of *some* plausible world state and (2) provide accurate coverage across

the *entire* distribution of the world that is possible from its conditioning signal. In this section, we examine these two facets by studying $p(t|y)$: the distribution of dropping times possible from an object at coordinate $y$ *in the image plane*. To do this, we create a simulated dataset that has a much wider distribution $p(t|y)$ than our PSFT dataset. See Appendix F for more details on its construction.

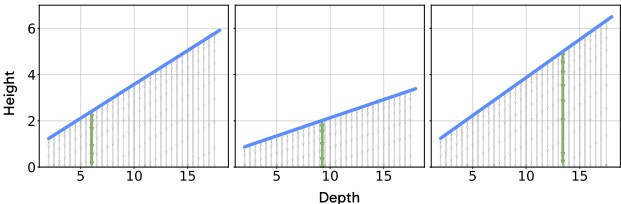

*Figure 10.* Examples of model trajectories lifted to 3D. The blue line represents the height of the camera ray passing through the bottom of the dropping object as a function of depth. The set of possible dropping trajectories at a given depth are depicted in gray. The lifted trajectory of the model is depicted in green.

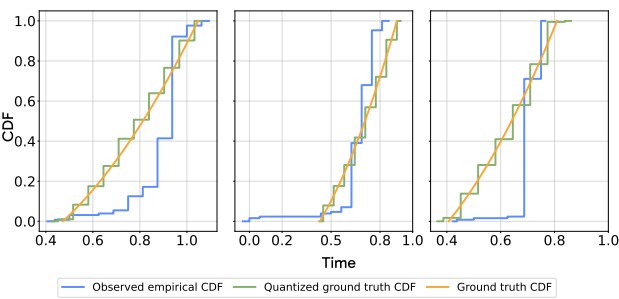

*Figure 11.* Visualizing $p(t|y)$ misalignment for different images. Green shows the ground-truth CDF, orange is the 32-frame quantized version, and blue is the empirical CDF of 128 different samples of dropping times from the model.

**Testing (1): 3D faithfulness of trajectories.**

After training our model on this new dataset, we test whether its trajectories are consistent with a valid 3D world state. We first obtain an estimated dropping time from generated videos using the procedure described in Section 5.1. Using knowledge of the camera position, focal length, sensor width, and $y$, we can obtain an implied depth and height of the trajectory. We can then back-project the video trajectory to 3D and analyze whether they constitute physically accurate trajectories. We give further details about this process in Appendix G. As show in in Figure 10, we find that our model's lifted trajectories consistently align with the 3D trajectory at the height and depth implied by its dropping time, giving evidence that the model's visual outputs are faithful to some plausible real-world state.

**Testing (2): distributional alignment.**

Going beyond the level of individual trajectories, we study

the model's learned conditional distribution $p(t|y)$. We create 50 different initial images with differing values of $y$, generate 128 different videos from each, and estimate the dropping time in each video. Using the laws of gravity, the laws of perspective, and the assumption of uniform depth sampling in our dataset, we can analytically derive the probability $p(t|y)$ as

$$p(t|y) = \begin{cases} \frac{gt}{(Z_{\max} - Z_{\min})\beta}, & t_{\min} \leq t \leq t_{\max} \\ 0, & \text{otherwise} \end{cases} \quad (3)$$

where $\beta$ is a constant that depends on $f$, $y$ and the camera height. The derivation is given in Appendix E.

We then measure goodness-of-fit for each of the 50 experiments using the Kolmogorov–Smirnov (KS) test (Massey Jr, 1951). The null hypothesis of the KS test is that the two distributions being compared are equal, and we consider p-values less than 0.05 as evidence of misalignment. Since our measured times have limited precision and can only take 32 distinct values—due to estimating the contact frame—we approximate the ground truth $p(t|y)$ using a Monte Carlo method. We sample 1000 values from the ground truth distribution and then quantized them into 32 bins corresponding to their frame, which we use as ground truth observations in the KS test. We find that in all 50/50 cases, the p-value from the test is less than 0.05, which provides evidence that the model does *not* learn the correct distribution of dropping times. We visualize the misalignment between the empirical CDF of the model's in Figure 11.

*In summary, while our model's trajectories show promising tendencies to ground themselves to plausible 3D world states, the range of possible outputs from the model does not align with the ground truth distribution.*

# 6. Conclusion

This work studies post-training as an avenue for adapting adapting pre-trained video generator into world models. We introduce a post-training strategy that is highly effective in aligning our model. Our work raises interesting insights into the learned distributions of generative models. Qualitatively, large scale image or video generative models appear to excel at generating likely samples from the data distribution, but this alone does not imply that they match the data distribution well in its entirety. As long as a model is able to generate likely samples, global distributional misalignment is not necessarily a problem for content creation. However, this problem becomes critical for world models, where alignment across the entire distribution is necessary for faithful world simulation. The insights revealed by our study, made possible by our constrained and tractable setting, indicate that although post-training improves per-sample accuracy, general distributional alignment remains unsolved.

## Impact Statement

This paper aims to explore physics post-training for video diffusion models. We release a benchmark for image-to-video task, focusing on a simple yet fundamental scenario: object free-fall. Building on powerful video generation model Open-Sora(Zheng et al., 2024), we introduce Physics Supervised Fine-tuning (PSFT) and Object Reward Optimization (ORO) to effectively introduce and enhance dropping behavior in the model. There are many potential societal consequences of our work, none which we feel must be specifically highlighted here.

## Acknowledgment

We thank Boyang Zheng, Srivats Poddar, Ellis Brown, Shengbang Tong, Shusheng Yang, Jihan Yang, Daohan Lu, Anjali Gupta and Ziteng Wang for their help with data collection. We thank Jiraphon Yenphraphai for valuable assistance in setting up our simulation code. We thank Runway and Kling AI for providing API credit. SX also acknowledges support from Intel AI SRS, Korean AI Research Hub, Open Path AI Foundation, Amazon Research Award, Google TRC program, and NSF Award IIS-2443404.

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

## A. Discussion of Image-to-Video setting.

We note that our choice of single-image input, as opposed to multi-frame input, comes with some trade-offs. We choose the image-to-video setting because it is widely supported among many different models, allowing us to make effective comparisons across the current state-of-the-art. However, only conditioning on a single frame introduces significant ambiguity. Due to the loss of information caused by projecting the 3D world through perspective, it may not be possible to directly infer the size of the object or its height. In practice, we find our metrics are still reliable signals of task success, but we still study the problem of ambiguity more extensively in Section 5.2.

## B. Metric details.

We propose three metrics to assess the accuracy of trajectories, shape fidelity, and object permanence. Each of our metrics compare frames from the ground-truth video with the generated video. Because different models can operate at different fps, we perform fps alignment as part of our evaluation process. To perform fps alignment, we map each frame index of the generated videos to the ground truth using $f_{\text{gen}}$ and $f_{\text{gt}}$, where $f_{\text{gen}}$ and $f_{\text{gt}}$ are the fps of generated video and ground truth respectively. For $i$-th frame in the generated video, we find the corresponding aligned frame index $j$ in the ground truth video:

$$j = \text{round}(i \cdot \frac{f_{\text{gen}}}{f_{\text{gt}}}) \tag{4}$$

Through fps alignment, we downsample the ground truth video to match the frame number of the generated video. We denote the downsampled ground truth as $\{I_i^{\text{gt}}\}_{i=1}^N$ and the generated video as $\{I_i^{\text{gen}}\}_{i=1}^N$, where $N$ is the number of frames in the generated video.

**Trajectory L2.** For each frame in both the generated video and ground truth, we calculate the centroid of the masked region. We then compute $L_2$ distance between the centroids of corresponding frames:

$$L_2 = \frac{1}{N} \sum_{i=1}^N \|C_i^{\text{gen}} - C_i^{\text{gt}}\|_2 \tag{5}$$

where $C_i^{\text{gen}}, C_i^{\text{gt}} \in \mathbb{R}^2$ are the centroids of the dropping object in the $i$-th frame of generated video and the ground truth respectively.

**Chamfer Distance (CD).** To assess the shape fidelity of objects, we calculate the Chamfer Distance (CD) between the mask regions of the generated video and ground truth:

$$\text{CD} = \frac{1}{N} \sum_{i=1}^N \left( \frac{1}{|P_i|} \sum_{p \in P_i} \min_{q \in Q_i} \|p - q\|_2 + \frac{1}{|Q_i|} \sum_{q \in Q_i} \min_{p \in P_i} \|q - p\|_2 \right)$$

where $P_i = \{p_j\}_{j=1}^{|P_i|}$ and $Q_i = \{q_j\}_{j=1}^{|Q_i|}$ are the sets of mask points in the $i$-th frame of the generated video and ground truth respectively.

**Intersection over Union (IoU).** We use the Intersection over Union (IoU) metric to evaluate object permanence. IoU measures objects' degree of overlap between the generated video and ground truth. This is formulated as follows:

$$\text{IoU} = \frac{1}{|N|} \sum_{i=1}^N \frac{|M_i^{\text{gen}} \cap M_i^{\text{gt}}|}{|M_i^{\text{gen}} \cup M_i^{\text{gt}}|} \tag{6}$$

where $M_i^{\text{gen}}, M_i^{\text{gt}} \in \{0, 1\}^{H \times W}$ are binary segmentation masks of the falling object in the $i$-th frame of the generated and ground truth videos respectively.

**Time error.** When testing on videos generated in simulation, we can provide a timing error. From the dropping height $Y_0$ of the ground truth video, which we have access to from the simulator, we can derive $t_{\text{drop}} = \sqrt{Y_0 \frac{2}{g}}$. We then obtain a dropping time from the model's output by estimating the frame of impact as the first frame $F$ whose centroid velocity in the $y$ direction is negative. If $t_{\text{drop}}$ occurs in between $F$ and $F - 1$, then we define the time error $E_{\text{time}}$ as zero. Otherwise, we define the time error as

$$E_{\text{time}} = \min \left( \left| \frac{F - 1}{\text{fps}} - t_{\text{drop}} \right|, \left| \frac{F}{\text{fps}} - t_{\text{drop}} \right| \right). \tag{7}$$

# C. ORO implementation details.

In our setting, we do not cut the gradient after step $k$ like VADER. The gradient $\nabla_\theta R(x_0', x_0)$ backpropagates through all diffusion timesteps and update the model weights $\theta$:

$$\nabla_\theta(R(x_0', x_0)) = \sum_{t=0}^{T} \frac{\partial R(x_0', x_0)}{\partial x_t} \cdot \frac{\partial x_t}{\partial \theta} \tag{8}$$

where T is the total diffusion timesteps.

**Segmentation Reward.** We utilize SAM 2 (Ravi et al., 2024) to generate segmentation masks across frames for generated video:

$$M^{\text{gen}} = \text{SAM-2}(x_0) \tag{9}$$

where $M^{\text{gen}}$ denotes the masks of the falling object in the generated video. We obtain ground truth masks $M^{\text{gt}}$ using Kubric (Greff et al., 2022). To avoid non-differentiable reward, we use $\text{Sigmoid}$ to normalize mask logits of generated video instead of converting them to binary masks. We use IoU between $M^{\text{gen}}$ and $M^{\text{gt}}$ as reward function:

$$R(x_0', x_0) = \text{IoU}(M^{\text{gen}}, M^{\text{gt}}) \tag{10}$$

Maximizing objective 1 is equivalent to minimizing the following objective:

$$J(\theta) = \mathbb{E}_{(x_0,c)\sim\mathcal{D},x_0'\sim p_\theta(x_0'|c)}[1 - \text{IoU}(M^{\text{gen}}, M^{\text{gt}})] \tag{11}$$

This objective constrains the position and shape of the generated object in the video, encouraging a greater intersection with the object region in the ground truth video. The model learns to generate more accurate object positions and shapes through training with this objective.

**Optical Flow Reward.** We utilize RAFT (Teed & Deng, 2020) to generate optical flow for both generated videos and ground truth:

$$\begin{aligned} V^{\text{gen}} &= \text{RAFT}(x_0') \\ V^{\text{gt}} &= \text{RAFT}(x_0) \end{aligned} \tag{12}$$

where $V^{\text{gen}}, V^{\text{gt}}$ denote the optical flows of generated videos and ground truth. We define the reward as follows:

$$R(x_0', x_0) = -|V^{\text{gen}} - V^{\text{gt}}| \tag{13}$$

Maximizing objective 1 is equivalent to minimizing the following objective:

$$J(\theta) = \mathbb{E}_{(x_0,c)\sim\mathcal{D},x_0'\sim p_\theta(x_0'|c)}[|V^{\text{gen}} - V^{\text{gt}}|] \tag{14}$$

This objective constrains the motion of the generated object in the video. The model learns to generate more accurate physical motion through training with this objective.

**Depth Reward.** We utilize Depth-Anything-V2 (Yang et al., 2024a) to generate optical depth maps for both generated videos and ground truth:

$$\begin{aligned} D^{\text{gen}} &= \text{Depth-Anything-V2}(x_0') \\ D^{\text{gt}} &= \text{Depth-Anything-V2}(x_0) \end{aligned} \tag{15}$$

where $D^{\text{gen}}, D^{\text{gt}}$ denote the depth maps of generated videos and ground truth. We define the reward as follows:

$$R(x_0', x_0) = -|D^{\text{gen}} - D^{\text{gt}}| \tag{16}$$

Maximizing objective 1 is equivalent to minimizing the following objective:

$$J(\theta) = \mathbb{E}_{(x_0,c)\sim\mathcal{D},x_0'\sim p_\theta(x_0'|c)}[|D^{\text{gen}} - D^{\text{gt}}|] \tag{17}$$

This objective constrains the 3d motion of the generated object in the video. The model learns to generate more accurate 3d physical motion through training with this objective.

## D. Coordinate system

We give a visualization of the coordinate system used in this paper in Figure 12. To compute $y$, we first leverage a segmentation map and find pixel row index that is just below the object. Once this row index is found, $y$ can easily be computed from the camera position, camera sensor size, and image resolution. We note that because our camera is assumed to be in perspective with the $XY$ plane, we can ignore $X$ and $x$ (not shown in figure) in our analyses in Section 5.1 and Section 5.2.

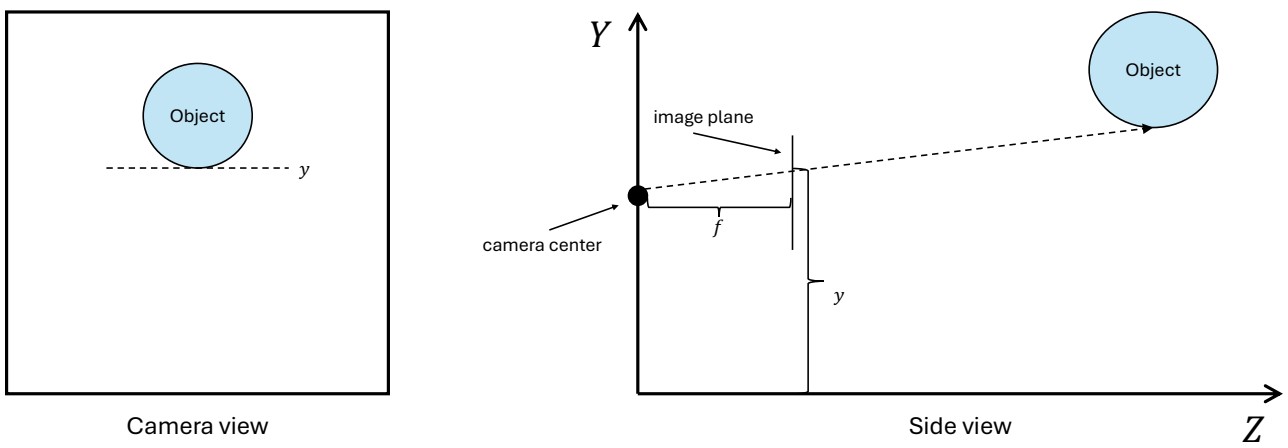

Figure 12. A visualization of the coordinate system used in this paper (not to scale). The image plane height of the object is denoted as $y$, its actual height in 3D as $Y$, and its depth as $Z$. The camera focal length is denoted as $f$.

## E. Derivation of $p(t|y)$

In our dataset construction, we assume a uniform distribution for $Z$, where $Z \sim \mathcal{U}(Z_{\min}, Z_{\max})$, where $Z_{\min} = 2$ and $Z_{\max} = 18$. As shown in Figure 12, the dropping height $Y$ is a linear function of $Z$, i.e. $Y = y + \beta Z$ for the slope $\beta$ that can be computed from $y$, $f$, the sensor size, and the camera height. This means we can solve for dropping time as $t = \sqrt{\frac{2}{g}Y} = \sqrt{\frac{2}{g}(y + \beta Z)}$. Applying the transformation rule for probability density yields

$$p(t|y) = \begin{cases} \frac{gt}{(Z_{\max} - Z_{\min})\beta}, & t_{\min} \leq t \leq t_{\max} \\ 0, & \text{otherwise} \end{cases} \tag{18}$$

where $t_{\min} = \sqrt{\frac{2}{g}(y + \beta Z_{\min})}$ and $t_{\max} = \sqrt{\frac{2}{g}(y + \beta Z_{\max})}$. Plugging in $Z_{\min} = 2$ and $Z_{\max} = 18$ yields Equation (3).

## F. Ambiguous dataset

We introduce a new dataset for distributional analysis that broadens $p(t|y)$, in contrast to the PSFT dataset, which prioritizes realism and has a narrower distribution due to limited object depth variability. To create a dataset with $p(t|y)$ that is sufficiently diverse for meaningful analysis, we first set up the initial scenes as before, but then apply an augmentation where a new depth values is sampled uniformly from $[2, 18]$ and the object is scaled and translated such that it appears the same in the original image, as shown in Figure 9. For simplicity, we limit our scenes to a single dropping object with no other objects on the ground. We also disable shadows, preventing the model from using them as cues to infer depth and height. Our dataset contains 5k samples consisting of 1k unique initial scenes each containing 5 different trajectories produced by the augmentation.

## G. Lifting trajectories to 3D

To lift trajectories to 3D, we first estimate $t_{\text{drop}}$ as described in Section 5.1. Using SAM2 to estimate object masks in the generated video, we can obtain a trajectory of the bottom of the object which we denote as $y_0, y_1, \ldots, y_N$ where

$N = t_{\text{drop}} \times$ fps. From $t_{\text{drop}}$, we can solve for an implied depth $Z = \frac{\frac{1}{2}gt^2 - y}{\beta}$. We then compute the lifted 3D trajectory as $y_i \mapsto y_i + \beta Z$

## H. PisaBench Details

In this section, we discuss the details of our data collection pipeline and annotations. We present more examples of real-world videos and corresponding annotations in Figure 13.

### H.1. Data Collection Pipeline

**Collecting Real World Videos.** We enlist approximately 15 volunteers to participate in the data collection process. We hand out a tripod, tape, and invisible wire for each volunteer. To ensure the quality, diversity, and minimize the ambiguity introduced by the environments, volunteers are provided with detailed guidelines. The key points of the data collection guidelines are shown in Table 3.

**Raw videos processing.** For the collected raw videos, we cut each video into multiple clips and crop their sizes. For each video clip, we annotate its starting position in the original long video and ensure that the duration of each segment does not exceed 12 seconds. Regarding the sizes of the videos, we manually crop each video to an aspect ratio of $1:1$, ensuring that the falling objects remain fully visible within the frame during the cropping process. The processing interface is shown in Figure 14.

### H.2. Annotation Details

We present our annotation details Figure 15. For video captions, we present the word cloud figure in (a). For segmentation masks, we annotate all objects in the first frame using positive and negative points, which are then propagated across frames using the SAM 2 (Ravi et al., 2024) model to produce segmentation masks for all objects throughout the video. The annotation interface is shown in (b).

In addition to providing the annotated caption "{object description} falls.", we also add information to inform off-the-shelf models of the task's context as much as possible. To further enhance task comprehension, we append an additional description "A video that conforms to the laws of physics." We also employ negative prompts "no camera motion" and "no slow-motion" to ensure environmental stability and impose constraints on the generated videos. These prompts explicitly instruct the models to avoid including camera motion or any non-real-time object motion, thereby maintaining consistency with real-world physics.

## I. Inference Details

We present the inference configurations of each closed or open model we evaluate in Table 4. For models that do not support generating videos with 1:1 aspect ratio, we pad initial frames with black borders to the resolution supported by these models, and finally remove the black borders from the generated videos.

## J. More Qualitative Examples

We present more qualitative examples in Figure 16 - Figure 22. Although in some showcases, models can roughly predict the downward trend, models still struggle to predict plausible shape and motion. The defects in the models can be mainly attributed to the following aspects:

- Trajectory correctness: in most videos, models fail to predict even the basic falling trajectory of objects, as shown in Figure 19 (a), despite this being highly intuitive for humans. Even in cases where the falling trajectory is roughly correctly predicted, the models still struggle to accurately predict subsequent events, such as collisions, as illustrated in Figure 16 (f).

- Object consistency: in many generated videos, object consistency is poor. Models struggle to infer the appearance of objects from multiple viewpoints in a physically plausible manner, resulting in unnatural appearances, as shown in Figure 16 (a). Additionally, models perform poorly in maintaining object permanence, causing objects to appear blurry, as illustrated in Figure 20 (f). Furthermore, models sometimes introduce new objects into the video, as depicted in

Figure 20 (e).

- Scene consistency: models struggle to maintain scene consistency, leading to abrupt transitions in many videos. These sudden changes make videos appear unnatural, as shown in Figure 18 (f).

## K. Simulated Adaption Details

We use the Kubric (Greff et al., 2022) simulation and rendering engine for creating our simulated videos. Kubric uses PyBullet (Coumans et al., 2010) for running physics simulations and Blender (Community, 2018) for rendering. We set the simulation rate to 240 steps per second and render 2-second videos at 16 fps, resulting in 32 frames per video. Each scene consists of objects from the Google Scanned Objects (GSO) dataset (Downs et al., 2022) and uses environmental lighting from HDRI maps provided by Kubric. We use 930 objects and 458 HDRI maps for training and 103 objects and 51 HDRI maps for testing.

For each video, we randomly choose 1-6 objects to drop. These objects are placed at a height uniformly sampled from 0.5m to 1.5m. Below each of these objects, a possibly empty pile of up to 4 objects spawns beneath to create collisions. The objects are placed in a spawn region of size 2m × 2m.

The camera is initially positioned 1m behind this region, with its height varying uniformly between 0.4m and 0.6m. Once all objects are placed, the camera moves back in random increments until all objects are visible within the camera frame. The camera uses a focal length of 35 mm, a sensor width of 32mm, and an aspect ratio of $1 \times 1$.

## L. Limitations

In this work, we collect and manually annotate a dataset of 361 real-world videos and design three spatial metrics to evaluate the performance of state-of-the-art image-to-video (I2V) models in a fundamental physical scenario: free fall. Our metrics focus solely on spatial positional relationships, excluding object appearance attributes such as color. To enable more fine-grained evaluations of appearance characteristics, we aim to develop metrics based on Multimodal Large Language Models (MLLMs) or pixel-level analysis in future work.

Furthermore, we propose the PSFT and ORO methods to fine-tune the Open-Sora model (Zheng et al., 2024), improving its ability to generate physically plausible videos. Despite these improvements, certain limitations remain, specifically, the generation of blurry objects in some videos. We hope to address these challenges in future research by refining both the dataset and the fine-tuning strategies, aiming to produce videos that better maintain object visuals.

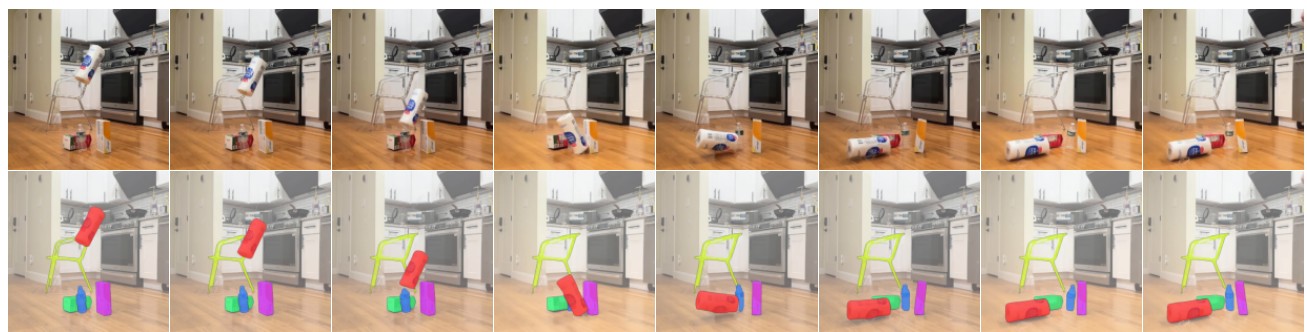

(a) A white paper roll falls.

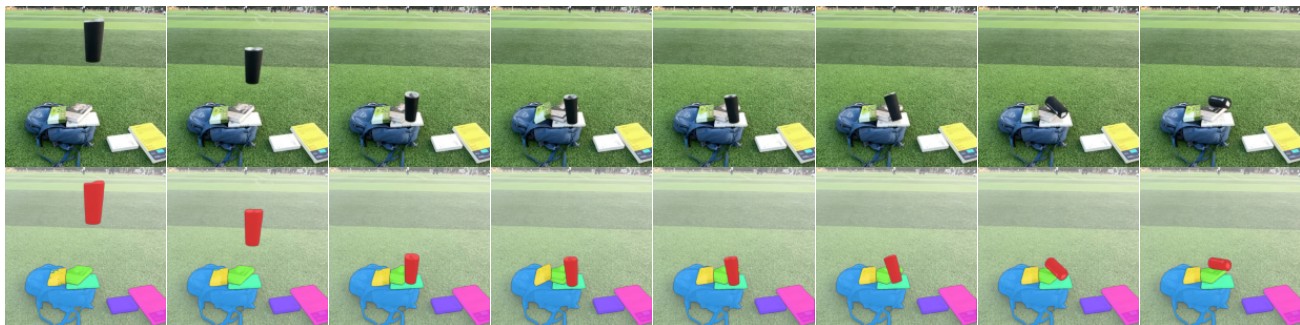

(c) A black bottle falls.

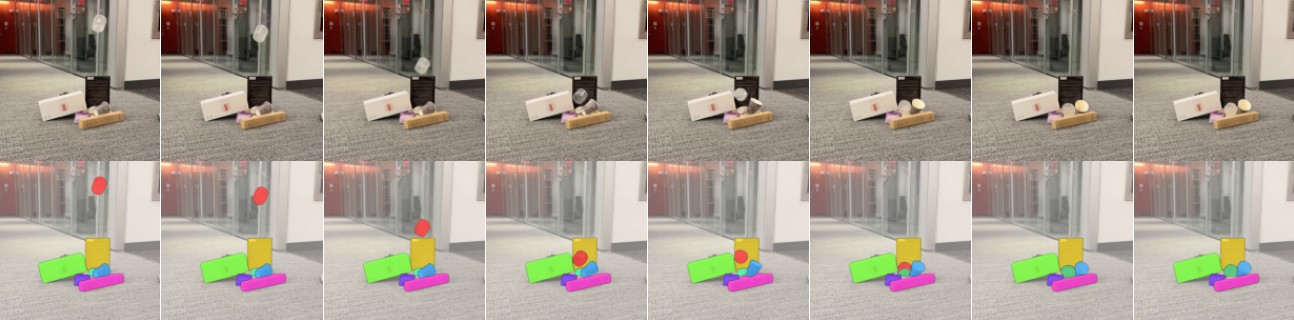

(b) A transparent bottle falls.

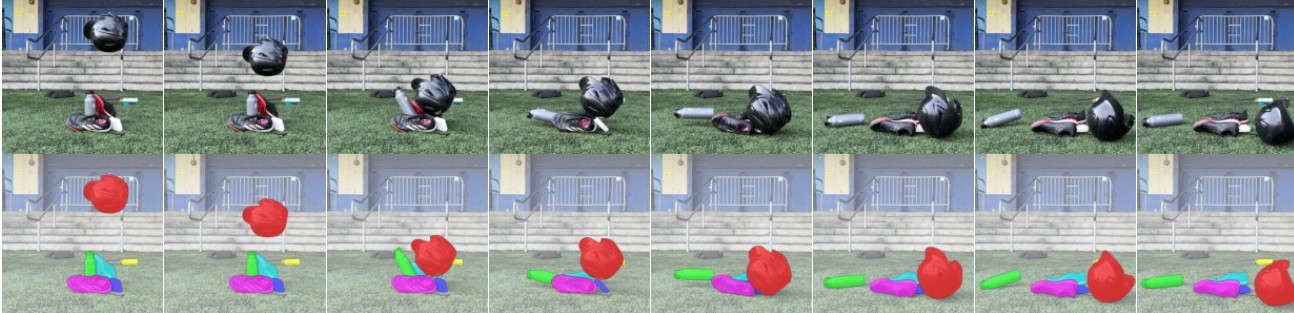

(d) A white bottle falls.

*Figure 13.* Examples of real world videos and annotations. We present video frames in the first row and mask annotations in the second row.

| Aspect | Requirements |
|---|---|
| Camera | <ul><li>The camera must be stabilized using a tripod.</li><li>The dropping object should remain visible throughout the entire fall.</li><li>The trajectory of the object should be sufficiently centered in the frame.</li><li>Ensure the slow-motion setting is configured to 120 fps.</li><li>Avoid a completely top-down perspective; the frame should include both the floor and the wall for spatial context.</li><li>It is acceptable to record one long video containing multiple drops at the same location.</li></ul> |
| Objects | <ul><li>Most objects should be rigid and non-deformable.</li><li>A limited number of flexible or deformable objects may be included, as such data is also valuable.</li></ul> |
| Dropping Procedure | <ul><li>Secure the object with a wire using tape, ensuring stability. Multiple tapings may be necessary for proper stabilization.</li><li>Visibility of the wire in the video is acceptable.</li><li>No body parts should appear in the frame. If this is challenging, consider having a partner monitor the camera or use screen-sharing software to view the camera feed on a laptop for uninterrupted framing.</li><li>Record videos in a horizontal orientation to simplify cropping and to help keep the frame free of unnecessary elements.</li><li>Use a short wire to enhance object stability.</li><li>The object should remain stationary before being dropped.</li></ul> |
| Scene Composition | <ul><li>Make the scenes dynamic and engaging. Include interactions with other objects, such as collisions or objects tipping over. Static objects should serve as active elements rather than mere background props.</li><li>Avoid filming in classroom or laboratory environments.</li><li>Include a variety of dropping heights.</li><li>Film in different environments, ensuring at least one setting is outside your apartment.</li><li>Minimize human shadows in the frame whenever possible.</li><li>Ensure good lighting and maintain strong contrast between the objects and the background.</li></ul> |

*Table 3.* Key points of real world videos collection guideline. We have detailed requirements for camera, objects, dropping procedure and scene composition to ensure the quality, diversity and minimize ambiguity introduced by environments.

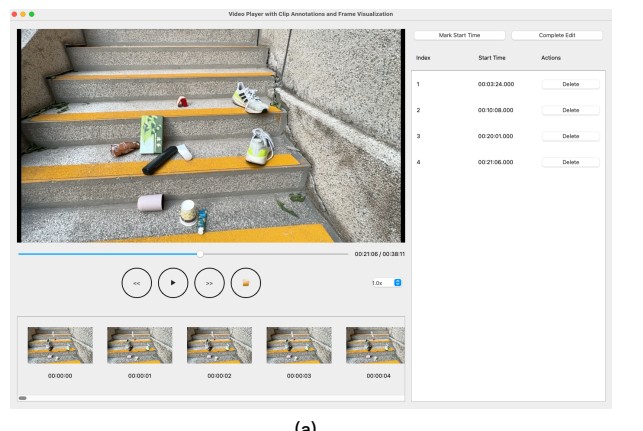

(a)

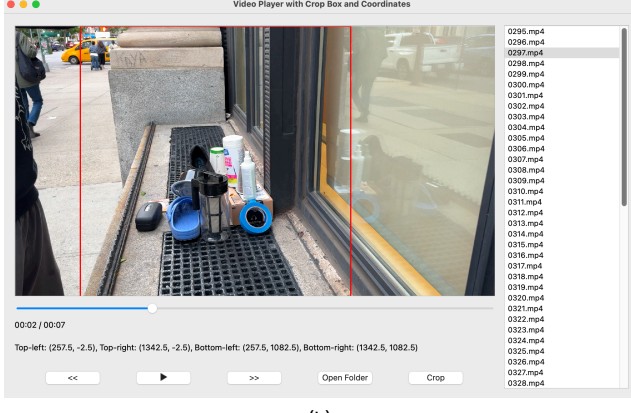

(b)

*Figure 14.* Video processing interface. (a) we annotate starting positions in the original long videos and clip them into multiple clips less than 12 seconds. (b) We drag the cropping box to crop the video size to an aspect ratio of 1:1.

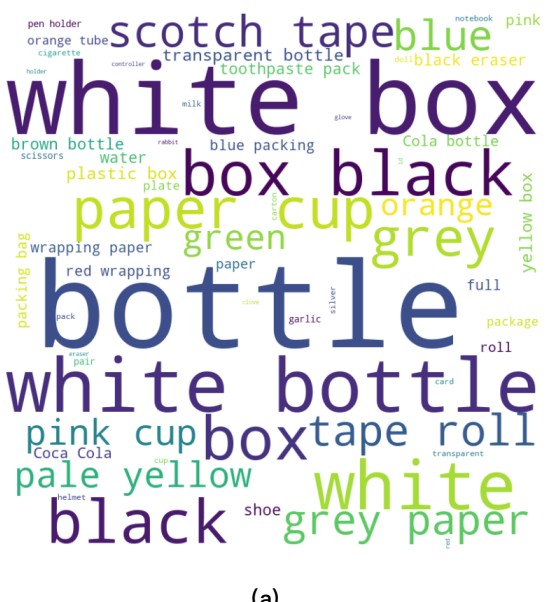

(a)

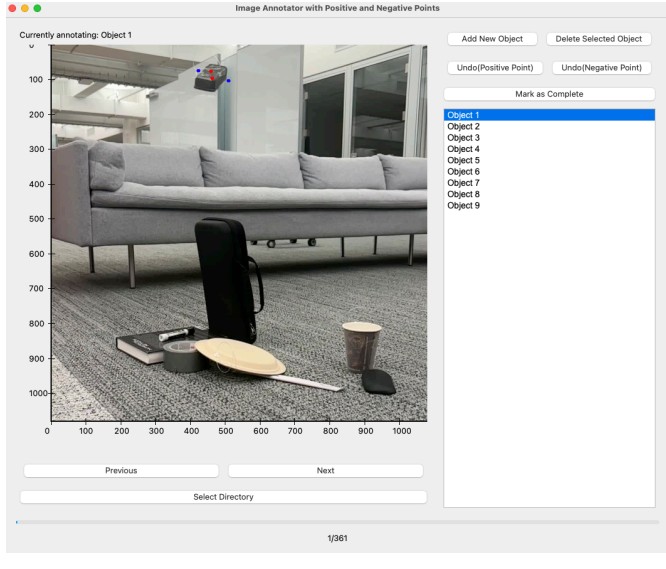

(b)

*Figure 15.* Annotation details of real world videos. (a) Word cloud of objects in video captions. Our videos contain a variety of daily life objects. (b) Interface for annotating positive and negative points in the first frame. Red and blue dots indicate positive and negative points respectively. We annotate all objects in the midair and ground.

| | Model | Resolution | Number of Frames | FPS | Guidance Scale | Sampling Steps | Noise Scheduler |
|---|---|---|---|---|---|---|---|
| Closed | Sora | $720 \times 720$ | 150 | 30 | - | - | - |
| | Kling-V1.5 | $960 \times 960$ | 150 | 30 | 1.0 | - | - |
| | Kling-V1 | $960 \times 960$ | 150 | 30 | 1.0 | - | - |
| | Runway Gen3 | $1280 \times 768$ | 156 | 30 | - | - | - |
| Open | CogVideoX-5B-I2V | $720 \times 480$ | 48 | 8 | 6.0 | 50 | DDIM |
| | DynamiCrafter | $512 \times 320$ | 90 | 30 | 0.7 | 50 | DDIM |
| | Pyramid-Flow | $1280 \times 768$ | 120 | 24 | 4.0 | 10 | EulerDiscrete |
| | Open-Sora | $512 \times 512$ | 90 | 30 | 7.0 | 30 | RFLOW |

*Table 4.* Inference details for models we evaluate, where "-" indicates the information is not available.

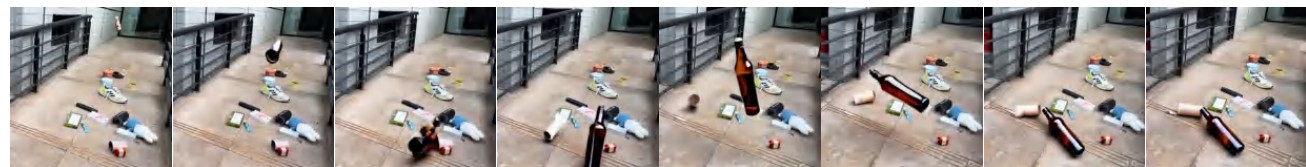

(a) A brown bottle falls.

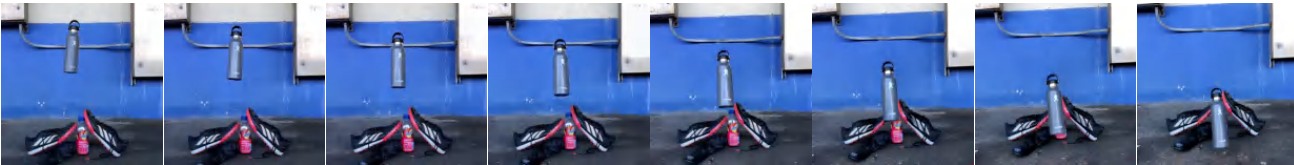

(b) A grey bottle falls.

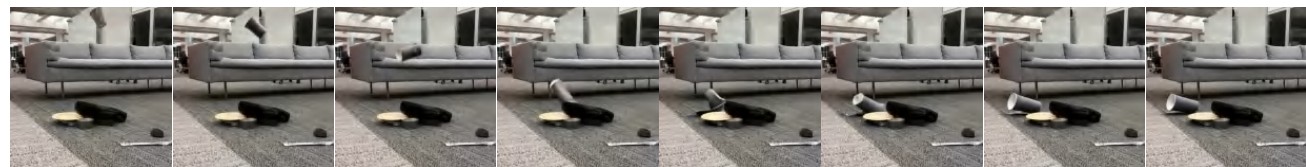

(c) A grey paper cup falls.

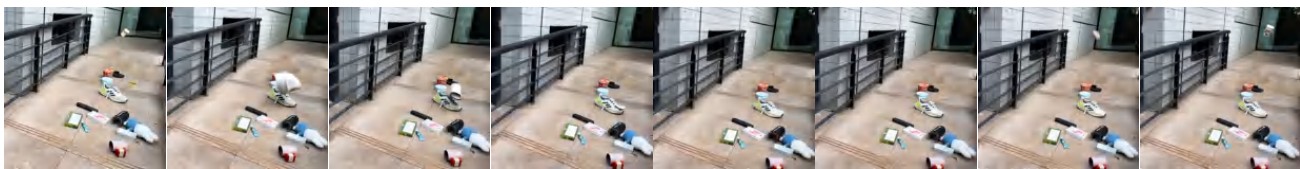

(d) A paper cup falls.

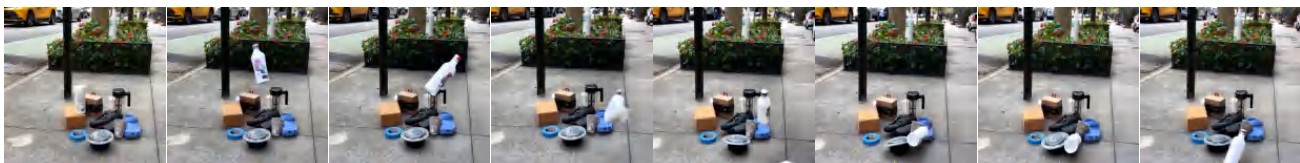

(e) A white bottle falls.

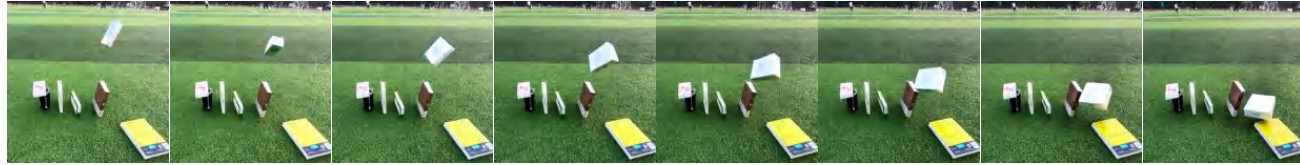

(f) A white box falls.

*Figure 16.* Qualitative examples of Kling-V1 (Kuaishou, 2024). In (a) (b) (c) (f), objects have a tendency to fall. (b) (c) are roughly consistent with the laws of physics. In (a) (f), the shape of the object does not match the first frame. In (d), the paper cup is suspended in midair. In (e), new object is introduced. In (e), the model fails to correctly predict the collision that occurs after the white box falls and the chain of events that follows.

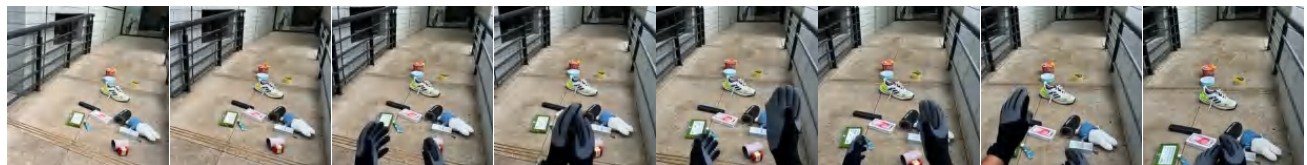

(a) A black and grey glove falls.

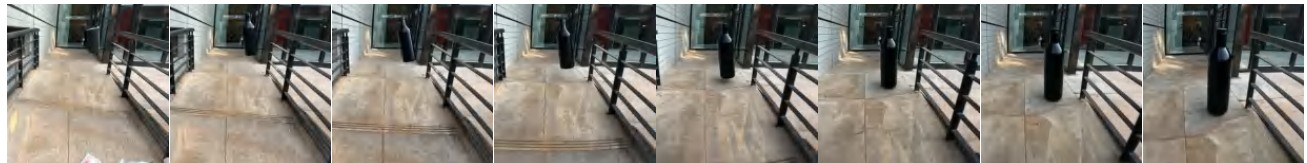

(b) A black bottle falls.

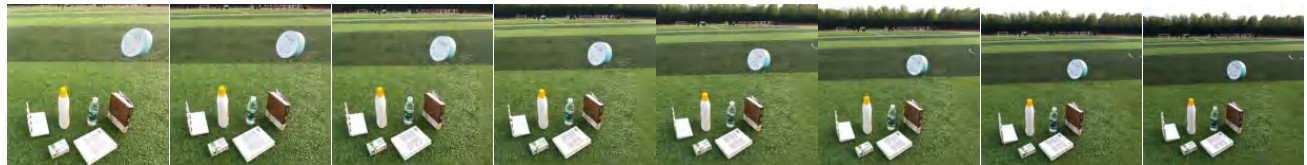

(c) A blue and white box falls.

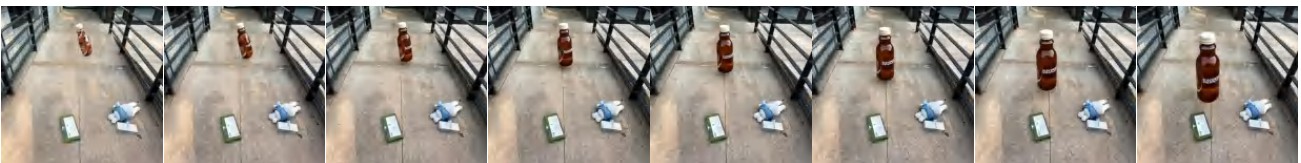

(d) A brown bottle falls.

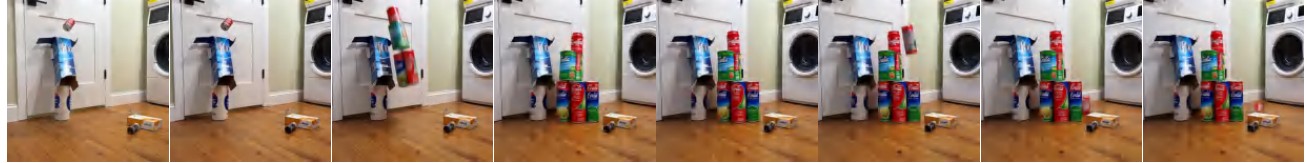

(e) A Coca–Cola can falls.

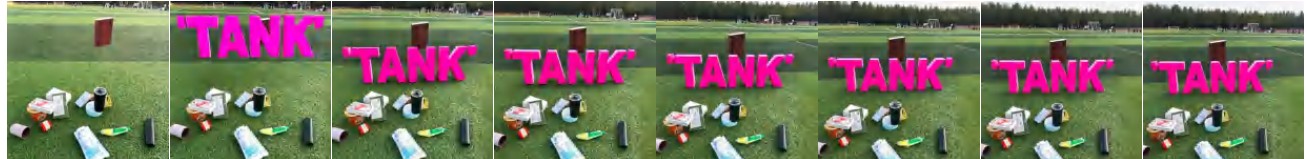

(f) A pink box falls.

*Figure 17.* Qualitative examples of Runway Gen3 (Runway, 2024). In (b) (e), objects have a tendency to fall. In (a) (e) (f), new objects are introduced. In (b) (d), the shape of the object does not match the first frame. In (c), the box is suspended in midair.

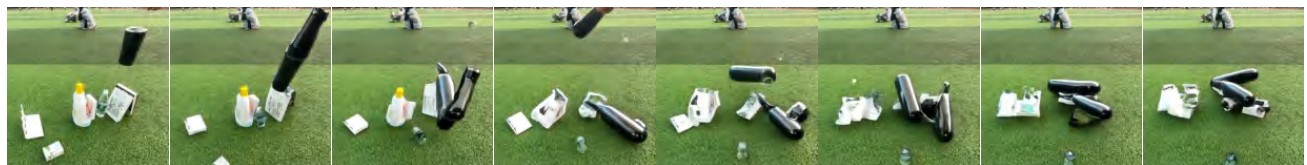

(a) A black bottle falls.

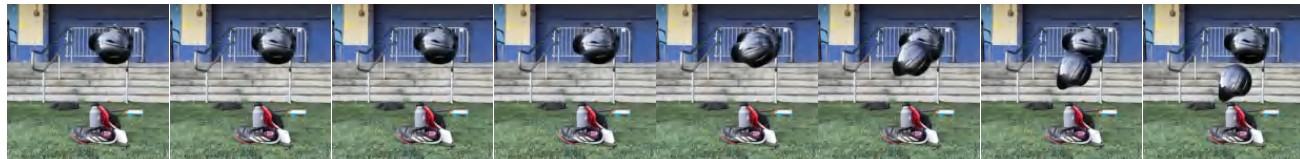

(b) A black helmet falls.

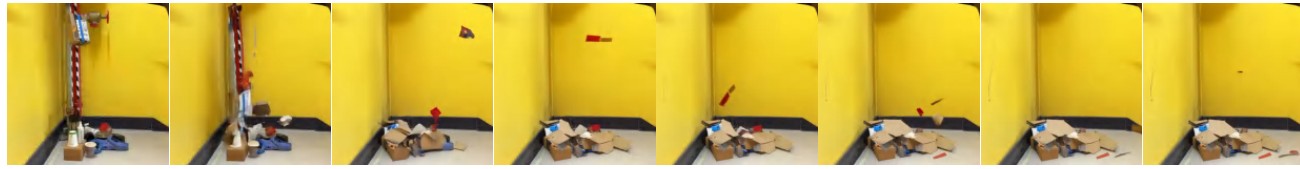

(c) A paper box falls.

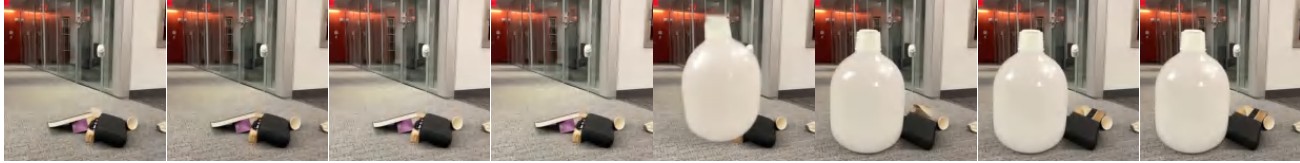

(d) A white bottle falls.

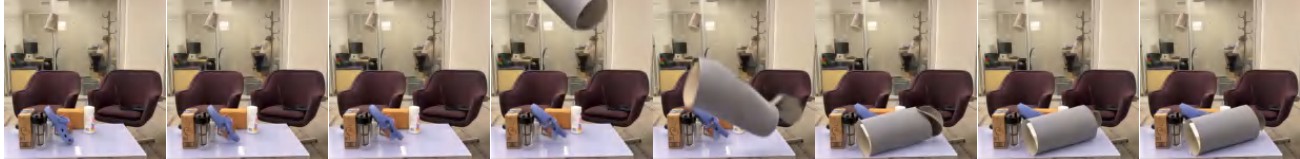

(e) A grey paper cup falls.

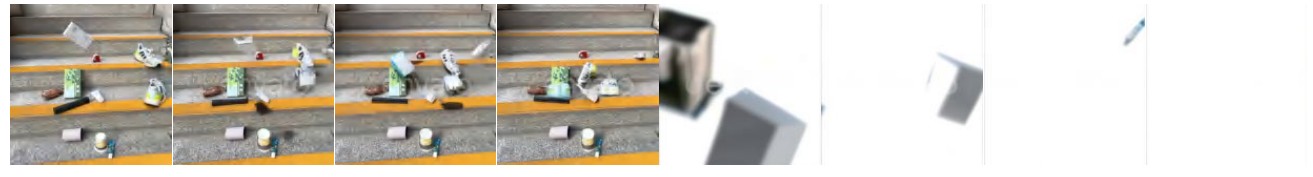

(f) A white box falls.

*Figure 18.* Qualitative examples of CogVideoX-5B-I2V (Yang et al., 2024c). In (a) - (f), objects have a tendency to fall. However, in all the videos, there are violations of physics. In (a) (b), the objects are divided into two parts. In (c) (d) (e), the shape of the object does not match the first frame. In (c), the trajectory is not a vertical fall. In (f), scene changes suddenly, which does not match the first frame.

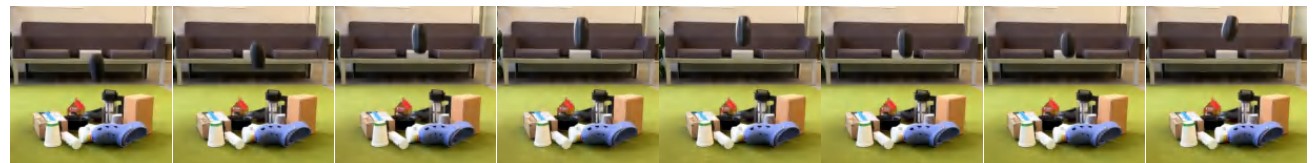

(a) A black box falls.

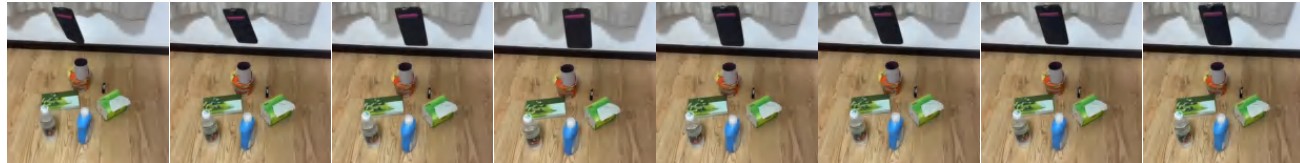

(b) A card holder falls.

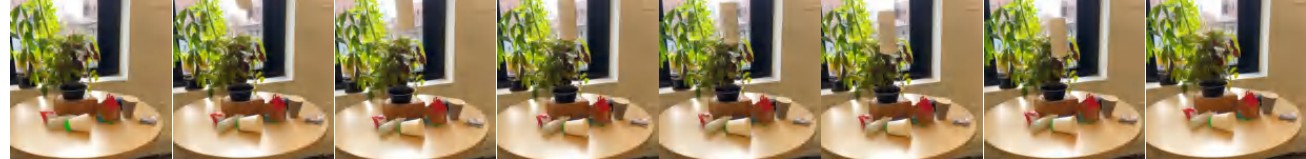

(c) A white bottle falls.

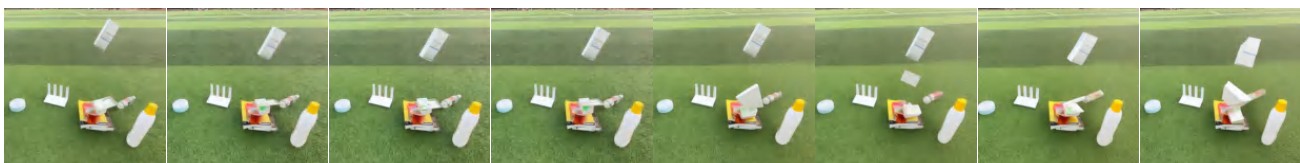

(d) A white box falls.

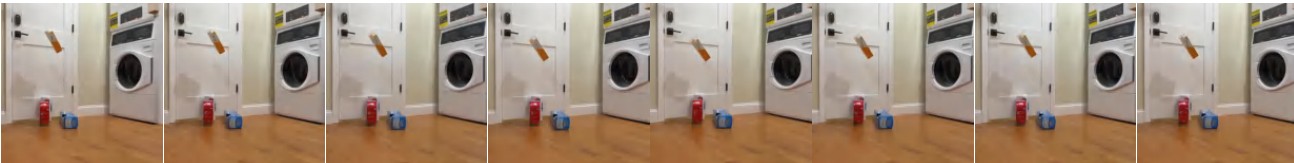

(e) An orange and white box falls.

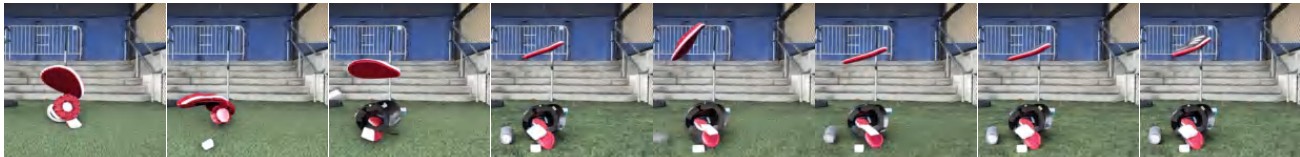

(f) A shoe falls.

*Figure 19.* Qualitative examples of DynamiCrafter (Xing et al., 2023). In all the videos, objects do not have a tendency to fall, suspended in the midair.

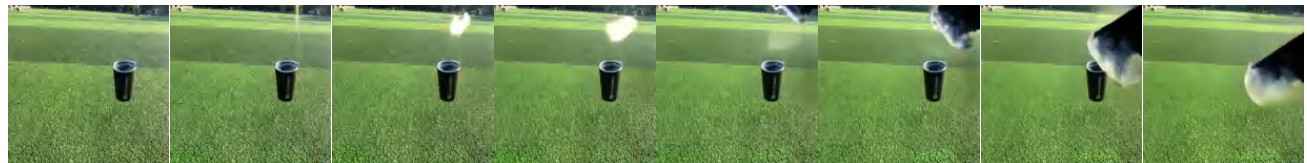

(a) A black bottle falls.

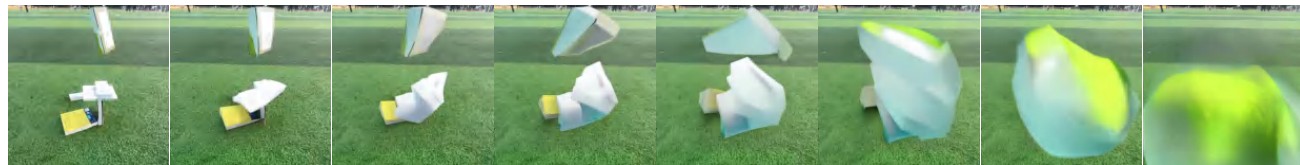

(b) A green and white box falls.

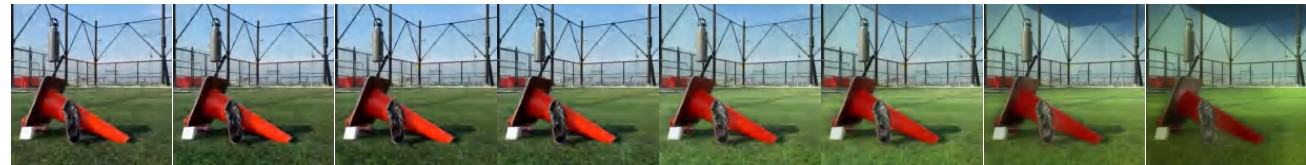

(c) A grey bottle falls.

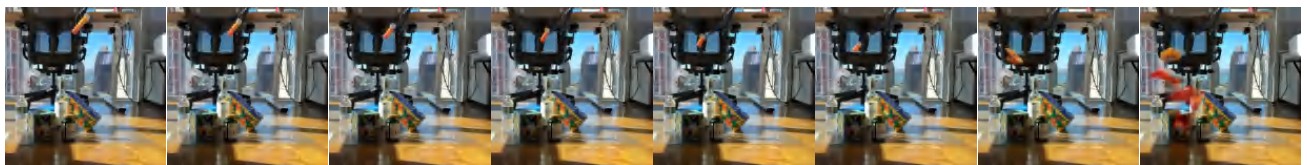

(d) An orange tube falls.

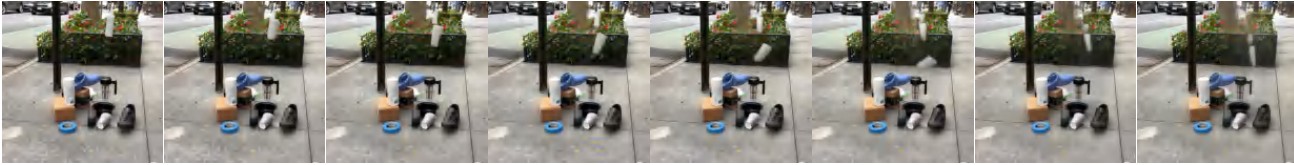

(e) A white bottle falls.

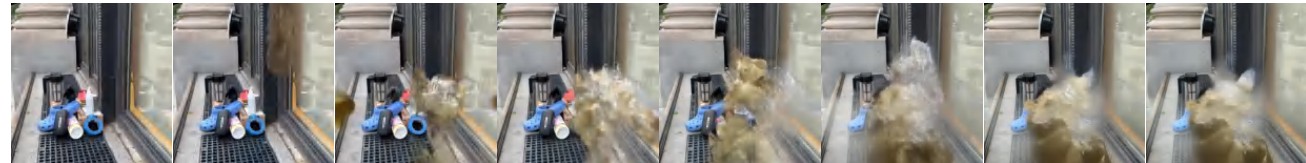

(f) A plastic box falls.

Figure 20. Qualitative examples of Pyramid-Flow (Jin et al., 2024). In (b) (d) (e), objects have a tendency to fall. In (a) (b) (e) (f), new objects are introduced. In (c), scene changes, which does not match the first frame.. In (d), the tube becomes blurry.

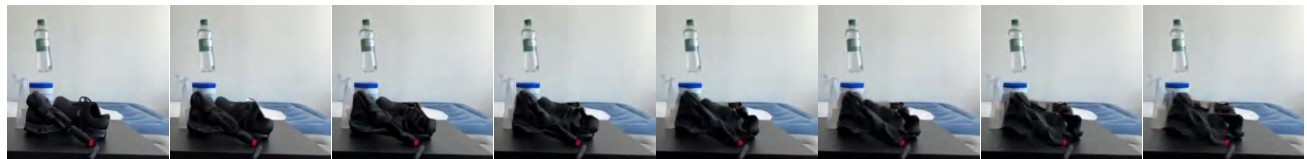

(a) A bottle full of water falls.

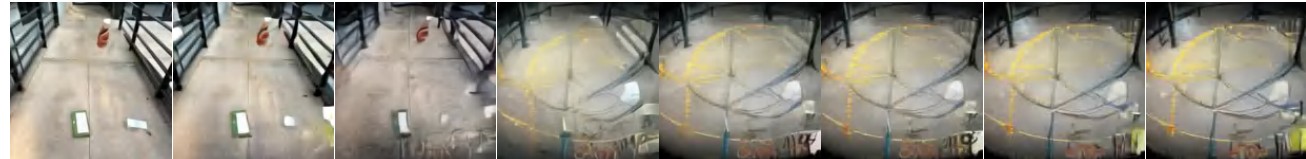

(b) A brown bottle falls.

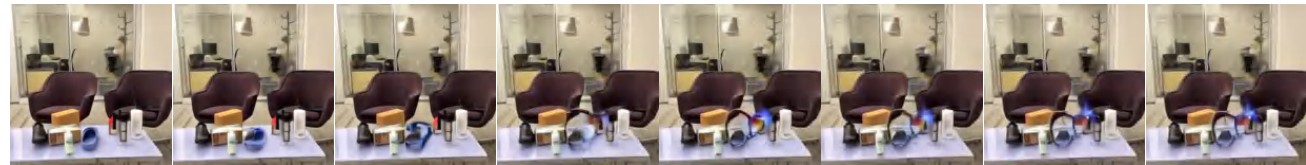

(c) A grey paper cup falls.

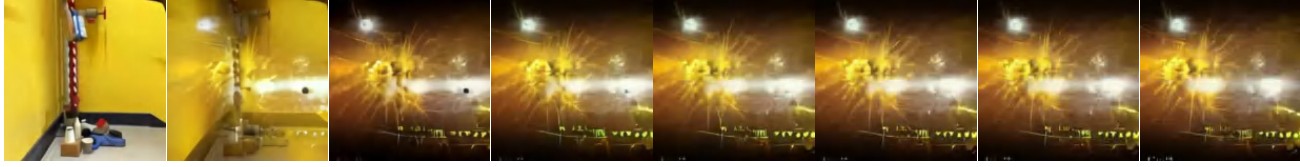

(d) A paper box falls.

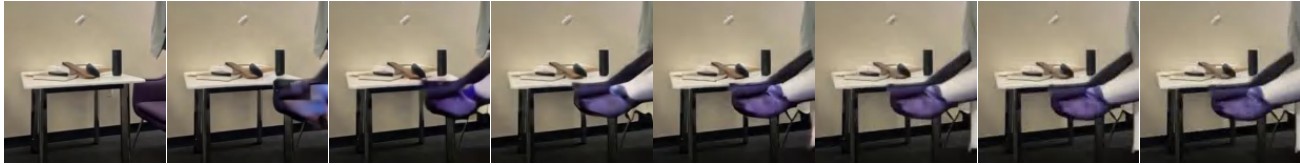

(e) A white bottle falls.

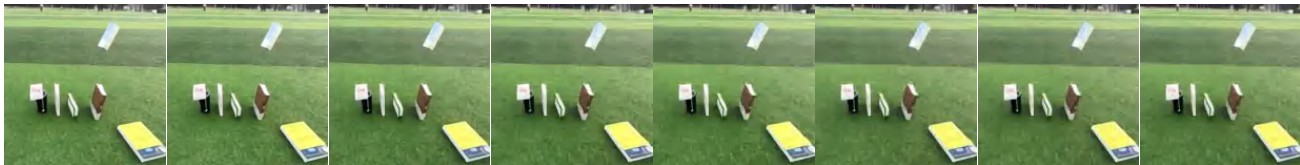

(f) A white box falls.

*Figure 21.* Qualitative examples of Open-Sora (Zheng et al., 2024). In all the videos, objects do not have a tendency to fall, suspended in the midair. In (b) (d), scene changes suddenly, which does not match the first frame. In (e), new object is introduced.

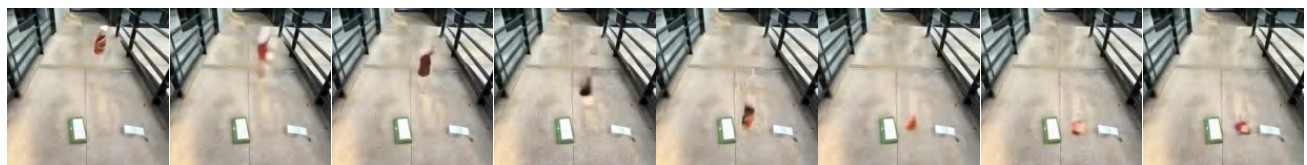

(a) A brown bottle falls.

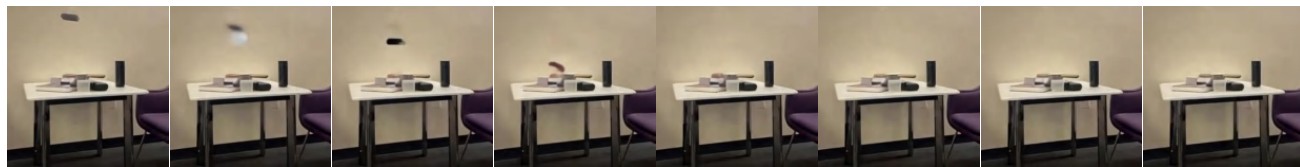

(b) A grey eraser falls.

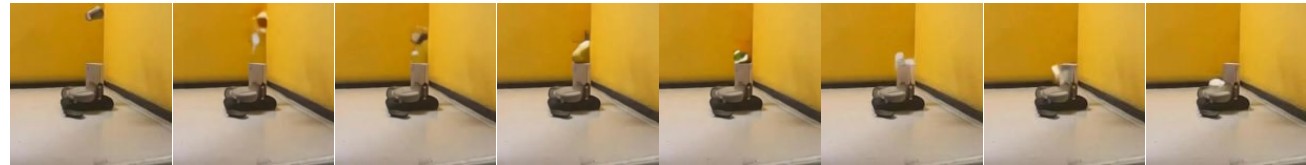

(c) A grey paper cup falls.

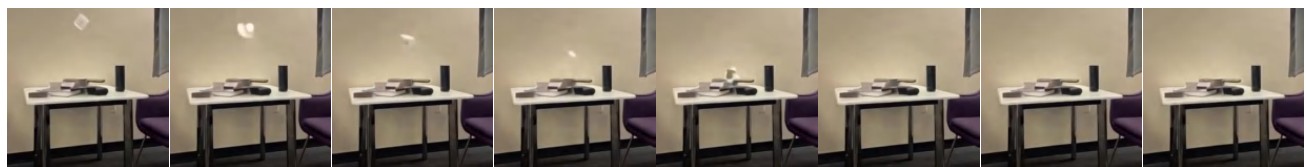

(d) A transparent bottle falls.

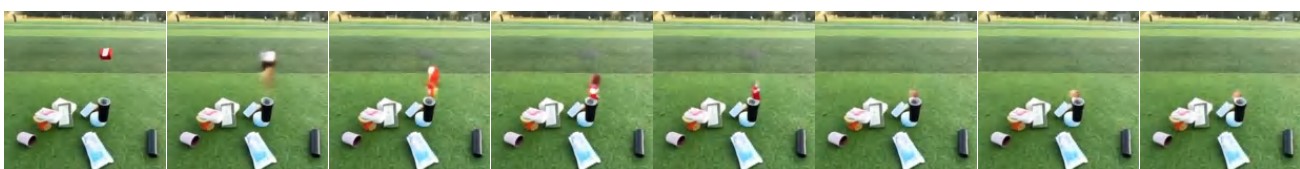

(e) A red wrapping paper falls.

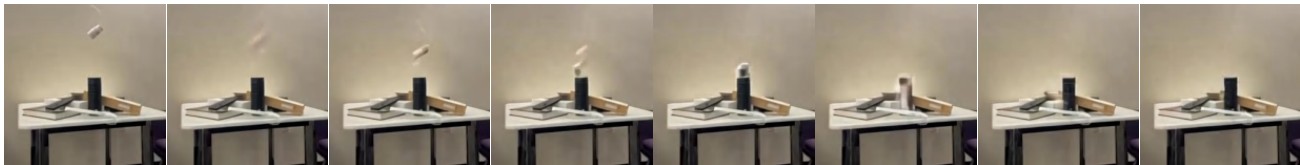

(f) A white bottle falls.

*Figure 22.* Qualitative examples of our method (Open-Sora + PSFT + ORO). In all the videos, objects have a tendency to fall. However, the consistency of objects is still insufficient. In some frames, objects become blurry. Objects sometimes disappear after collision.

