# OpenReview forum: "PISA Experiments: Exploring Physics Post-Training for Video Diffusion Models by Watching Stuff Drop"
_ICML.cc/2025/Conference — ICML 2025 poster_

### Official Review · Reviewer_WLEt · 2025-03-13

**Overall Recommendation:** 3

**Summary:**

Current large-scale pre-trained video generation models excel in content creation but are not suitable as physically accurate world simulators out of the box. Therefore, this paper introduces the PISA framework, providing diagnostic tools for assessing the physical modeling capabilities of video generation models. Experimental validation highlights the crucial role of post-training (fine-tuning and reward modeling) in enhancing physical accuracy.

**Claims And Evidence:**

The claims are supported by clear evidence.

**Essential References Not Discussed:**

N/A

**Experimental Designs Or Analyses:**

The authors designed various metrics to comprehensively reflect the degree of fit between the generated falling trajectories and the real trajectories. However, there are still some issues that require further research, such as the lack of an assessment of human subjective perception of physical realism. I believe this is crucial for the practical application of generated videos. It's difficult to intuitively reflect the degree of this realism with only the quantitative metrics provided in the paper.

**Methods And Evaluation Criteria:**

The methods and evaluation criteria make sense for the problem.

**Other Comments Or Suggestions:**

I have no further suggestions.

**Other Strengths And Weaknesses:**

The methods and benchmarks proposed in this paper are significant for enhancing the physical realism of generated videos. The authors' experiments found that while post-training can effectively improve physical realism, there are certain limitations, such as the incorrect distribution of dropping times, which has not been thoroughly discussed. Moreover, although the problem addressed in this paper is important and the proposed methods and benchmarks are valuable, I believe that this post-training approach struggles to fundamentally resolve the lack of physical realism. The authors' experiments also revealed generalization issues with post-training.

**Questions For Authors:**

I have no questions.

**Relation To Broader Scientific Literature:**

The methods and benchmarks proposed in this paper are significant for further investigating the key deficiencies in current video generation models.

**Theoretical Claims:**

The proofs are correct.

---

> ### Author Rebuttal · Authors · 2025-04-01
>
> We thank the reviewer for taking the time to review our work. We are extremely grateful for the feedback given, and below we address the main concerns raised.
>
> > **Q1:** Lack of human evaluation.
>
> This is a great point. While our work focused on quantitative evaluation, human evaluation is important too. To address this limitation, we asked four volunteers (none of them authors) to rank preferences in generation between our PSFT+ORO model and Sora. We showed 30 videos, 10 from our sim seen test set, 10 from our sim unseen test set, and 10 from our real world test set. For each video, we asked the anotators to select a preference for each of the three following questions:
>
> 1. Which video does better at following basic physics?
> 2. Which video has better visual realism?
> 3. Which video does a better job at preserving the state/identity of objects, i.e. not hallucinating or introducing implausible state changes?
>
> Overall, our model is prefered to Sora 90% of the time in physical accuracy, 56% of the time in visual realism, and 68% of the time in preserving object identity. The full results are shown [here](https://anonymous.4open.science/r/ICML-7650-Rebuttal-1645/7_human_eval.md).
>
>
> > **Q2:** Concern about distribution of dropping times.
>
> We believe that the distribution matching failure is an important insight for researchers developing world models. A goal of this paper is to make researchers aware of the strengths and weaknesses when post-training video models that struggle with basic physics. As such, we hope the reviewer agrees that negative results like this are important scientific findings that should be shared, even if they are limitations.
>
> > **Q3:** Struggle to fundamentally resolve the lack of physical realism.
>
> Our benchmark and human evaluation indicates our model to be more physically realistic than all other baselines, including frontier closed source models like Sora, on our dropping task. Our goal is not to solve physics as a whole, but rather to shed light on the post-training process that in our view will become an increasingly critical part of the world modeling stack. The strengths and limitations of this post-training process presented in our paper are valuable insights for future research.
>
> > **Q4:** Generalization issues.
>
> We evaluated our data on challenging and OOD settings in both real and simulated data. These videos include scenarios such as objects sliding down ramps, falling into containers, or domino-like setups. A summary of our relative improvement over the base OpenSora model is shown below.
>
> | Scenarios             | L2     | CD     | IoU     |
> | --------------------- | ------ | ------ | ------- |
> | Domino (Real)         | 47% | 54% | 42%  |
> | Ramp (Real)           | 41% | 47% | 18%  |
> | Stairs (Real)         | 25% | 20% | 103% |
> | Ramp (Simulated)      | 87% | 90% | 55%  |
> | Container (Simulated) | 75% | 67% | 3.7%   |
>
> Please see [here](https://anonymous.4open.science/r/ICML-7650-Rebuttal-1645/3_ood.md) for more information about the dataset construction and a full table breakdown across settings and comparisons with baselines.

---

> > ### Comment · Reviewer_WLEt · 2025-04-08
> >
> > Thank you for the authors' response, which has provided clearer answers to some of my questions.
> >
> > On one hand, the issue of physical rules in probabilistic generative video models represents a crucial yet challenging research direction with limited existing studies. This paper makes valuable exploratory contributions by proposing a feasible method to enhance physical realism, analyzing its effectiveness, and offering inspiration for subsequent research. However, on the other hand, the approach of "incorporating physics-specific data for post-training" appears overly simplistic and direct, lacking methodological innovation and analytical depth. Both its effectiveness and limitations seem easily foreseeable. Therefore, I maintain my original "weak accept" rating.
> >
> > From my perspective, the ultimate solution to this challenge should focus on model architecture and training paradigms rather than relying solely on data manipulation. For instance, why does the post-training process lead to significant divergence in performance between spatial trajectories and temporal trajectories in free-fall motion? Could this be related to the distinct modeling approaches for spatial and temporal dimensions in current video generation frameworks? I encourage the authors to conduct more in-depth investigations into these fundamental questions in future works.

---

### Official Review · Reviewer_hCaK · 2025-03-13

**Overall Recommendation:** 4

**Summary:**

The paper argues that the large-scale video generative modeling has enabled creative content generation but the accurate world modeling is missing. This is due to the complexities in the physical laws and perspectives that underpin the real-world videos. To solve this problem, the paper proposes to use targeted post-training. Specifically, the paper studies the potential of post-training in image-to-video generative models for freefalling objects under gravity.

The paper studies free falling because it is simple to simulate and evaluate to gain insights into the post-training stage. The experimental results suggest that: (a) the existing video generative models are quite bad at physically accurate object dropping, (b) simple finetuning of the video model on 1000s of examples fixes this problem significantly, (c) they consider further RL-tuning with multiple reward models and target multiple axes for physical improvement.

The paper also finds that finetuning struggles to generalize beyond the unseen depths and heights, and its struggle with trajectory distribution and dropping-time distribution. Overall, the paper performs insightful data collection and post-training that can serve as a useful data point for the community. I do have several comments on the work’s limitations and potential suggestions.

**Claims And Evidence:**

Yes

**Essential References Not Discussed:**

NA

**Experimental Designs Or Analyses:**

Yes

**Methods And Evaluation Criteria:**

Yes

**Other Comments Or Suggestions:**

- The evaluation metrics closely resemble those in the Physics-IQ paper [3], and proper attribution should be provided.

**Other Strengths And Weaknesses:**

**Strengths**
- The authors collect 361 real-world videos to assess model performance in real environments, which is crucial for quantifying the sim-to-real gap—an aspect missing from prior work [1].
- The entire framework is well-designed for the object dropping phenomenon -- having simulated videos for training the models and assessing them at different heights and depths for generalization study.
- The paper also goes beyond mere supervised finetuning and shows the ability to improve performance with further RL-tuning.

**Weaknesses**
- The study lacks a reliability measure with human evaluation; it is unclear whether humans prefer the finetuned model’s outputs over the generalist model’s.
- Figure 7 shows that the object dropping under gravity can be roughly broken down into two parts: straightline motion before impact and collision/rolling motion after impact. That figure suggests that the finetuned model has learned about behavior before motion pretty well but it is not clear about the later part. A good way to test this would be to breakdown the Table 1 finetuned model numbers into before impact and after impact.
- The study does not establish a strong conclusion on whether post-training improves world modeling in video models. The generalization capability appears limited, and experiments focus only on object falling—a relatively simple physical phenomenon. There is uncertainty about how the results extend to other physical behaviors, given the difficulty in acquiring simulated data for more complex scenarios.

[1] Section 5.3.2 in https://arxiv.org/abs/2501.03575v1
[2] https://arxiv.org/abs/2406.03520
[3] https://arxiv.org/abs/2501.09038

**Questions For Authors:**

Mentioned in the strengths and weaknesses

**Relation To Broader Scientific Literature:**

Mentioned in the summary, and strengths.

**Theoretical Claims:**

NA

---

> ### Author Rebuttal · Authors · 2025-04-01
>
> We thank the reviewer for taking the time to review our work. We are extremely grateful for the feedback given, and below we address the main concerns raised.
>
> > **Q1:** Lack of human evaluation.
>
> This is a great point. While our work focused on quantitative evaluation, human evaluation is important too. To address this limitation, we asked four volunteers (none of them authors) to rank preferences in generation between our PSFT+ORO model and Sora. We showed 30 videos, 10 from our sim seen test set, 10 from our sim unseen test set, and 10 from our real world test set. For each video, we asked the anotators to select a preference for each of the three following questions:
>
> 1. Which video does better at following basic physics?
> 2. Which video has better visual realism?
> 3. Which video does a better job at preserving the state/identity of objects, i.e. not hallucinating or introducing implausible state changes?
>
> Overall, our model is prefered to Sora 90% of the time in physical accuracy, 56% of the time in visual realism, and 68% of the time in preserving object identity. The full results are shown [here](https://anonymous.4open.science/r/ICML-7650-Rebuttal-1645/7_human_eval.md).
>
> > **Q2:** Evaluating before and after impact.
>
> We estimated the contact frame using the method described in Appendix B of the paper and ran our benchmark. Overall, we find that ORO most improves the results *after* the point of contact. This indicates that ORO is strongest at improving the most difficult aspects of physics modeling. The full set of evaluation tables can be found [here](https://anonymous.4open.science/r/ICML-7650-Rebuttal-1645/8_impact.md).
>
> > **Q3:** OOD generalization.
>
> We evaluated our data on challenging and OOD settings in both real and simulated data. These videos include scenarios such as objects sliding down ramps, falling into containers, or domino-like setups. A summary of our relative improvement over the base OpenSora model is shown below.
>
> | Scenarios             | L2     | CD     | IoU     |
> | --------------------- | ------ | ------ | ------- |
> | Domino (Real)         | 47% | 54% | 42%  |
> | Ramp (Real)           | 41% | 47% | 18%  |
> | Stairs (Real)         | 25% | 20% | 103% |
> | Ramp (Simulated)      | 87% | 90% | 55%  |
> | Container (Simulated) | 75% | 67% | 3.7%   |
>
> Please see [here](https://anonymous.4open.science/r/ICML-7650-Rebuttal-1645/3_ood.md) for more information about the dataset construction and a full table breakdown across settings and comparisons with baselines.
>
> > **Q4:** Attribution for Physics-IQ.
>
> Thank you for pointing this out. We did cite this paper in our related work section of our submission, though we did not point out the similarity between our concurrently developed metrics. We will definitely add an acknowledgement of this.
>
> Please let us know if you have any further questions or concerns. We are grateful that you feel our paper should be accepted, and if our response has sufficiently addressed the concerns you have mentioned, we would appreciate it if you consider raising your score further. Thank you very much again for taking the time to review our work!

---

### Official Review · Reviewer_ocR6 · 2025-03-16

**Overall Recommendation:** 3

**Summary:**

**Main Findings:**

- This paper addresses the physics-based task of modeling object freefall in video diffusion models, specifically formulated as follows: given an initial image of an object suspended midair, the goal is to generate a video depicting the object realistically falling, colliding with the ground, and potentially interacting with other objects.

- A new evaluation framework called PISA (Physics-Informed Simulation and Alignment) is proposed, including a video dataset. Results reveal that current state-of-the-art video generation models significantly struggle with accurately performing this fundamental physics task.

**Main Algorithmic/Conceptual Ideas:**

- The paper introduces a post-training method aimed at enhancing video generation models through Physics-Supervised Fine-Tuning (PSFT) and Object Reward Optimization (ORO).

**Main Results:**

- Evaluations using the proposed PISA framework demonstrate that existing SOTA video generation models have limited capabilities in accurately generating object freefall, highlighting weaknesses in their physical modeling abilities.

- While the proposed PSFT and ORO methods significantly enhance model performance on the benchmark task, their generalization to out-of-distribution (OOD) scenarios remains limited.

**Claims And Evidence:**

Yes, it is clear and convincing evidence.

**Essential References Not Discussed:**

No, the related work is solid.

**Experimental Designs Or Analyses:**

Yes, I have checked.

**Methods And Evaluation Criteria:**

Yes, the proposed methods and evaluation make sense for the target problem.

**Other Comments Or Suggestions:**

No additional comments.

**Other Strengths And Weaknesses:**

**Strengths:**

- The proposed PISA (Physics-Informed Simulation and Alignment) evaluation framework is well-motivated and thoughtfully designed. Experimental results clearly highlight current limitations in the physical modeling capabilities of state-of-the-art video generation models.

- The introduced PSFT and ORO methods demonstrate significant performance improvements on the PISA benchmark. The limitations in generalizing to out-of-distribution (OOD) scenarios are explicitly and effectively discussed.

**Weaknesses:**

- The scope of this paper is somewhat limited. Evaluating only object freefall addresses just a narrow aspect of the physical modeling capabilities of video generation models. The assessment would be more comprehensive and convincing if additional physical scenarios, such as collisions or diverse movement interactions (like in CLEVR dataset), were included.

- Post-training specifically on object freefall scenarios introduces a bias. Although PSFT and ORO effectively enhance performance on the PISA benchmark, the restricted variability in direction and speed during object freefall could allow for dataset-specific optimization or "hacking." This limitation is further corroborated by experimental evidence.

- The improvement methods, while effective, still demonstrate limited generalization to out-of-distribution (OOD) scenarios. This limitation should be addressed by introducing greater variability and complexity in training conditions.

**Questions For Authors:**

- If the video generation models become biased toward object freefall after post-training and lose their original capabilities? How might their performance be affected in scenarios involving collisions or movements in different directions (e.g., similar to CLEVR)? Additionally, how do these models perform on standard video generation benchmarks such as V-Bench?

- Regarding the GIF demonstrations provided in the supplementary materials, evaluating models based on higher-resolution object freefall experiments could yield more robust results. Lower-resolution demonstrations might limit the models' ability to accurately generalize semantic categories and effectively reason about object interactions, thereby posing additional challenges for evaluation.

**Relation To Broader Scientific Literature:**

Evaluating video generation models' capabilities in physical modeling is crucial for developing effective world models. Addressing this research problem is particularly significant for advancements in embodied AI.

**Theoretical Claims:**

Not applicable. No theoretical claim is proposed.

---

> ### Author Rebuttal · Authors · 2025-04-01
>
> We thank the reviewer for taking the time to review our work. We are extremely grateful for the feedback given, and below we address the main concerns raised.
>
> > **Q1:** Limited scope of the paper.
>
> Our goal in this paper is not to create a generalist state-of-the-art physics video model, but rather to treat the post-training process itself as the primary object of study. As shown in the paper, leading commercial models are unable to reliably simulate some of the most basic aspects of physics. As such, we believe that post-training is an overlooked aspect of video modeling that will soon become a mainstream research area, especially in world models/simulators for embodied AI applications.
>
> The simplicity of our task was deliberately chosen because it can serve as a probing mechanism for understanding aspects of the post-training process that would otherwise be overlooked in a more general setting. For example, we use the fact that our task can be analytically descibed with the laws of gravity and perspective to conduct the analysis in Sec. 5. We believe the distribution matching limitations are an important finding because if they are present in a setting as simple as ours, then the problem is likely to persist in more complex settings that are of interest to world model researchers. Hence this contribution is a highly valuable insight for future research looking to leverage video models as planners or simulators in embodied applications.
>
> > **Q2:** Bias in post-training.
>
> The problem of models overfitting/memorizing statistical patterns in the training data is ubiquitous across all areas of deep learning, even in the post-training of leading LLMs [1]. Our paper does not study solutions to this broader generalization problem, and the extent to which it is possible to solve is not clear either. In this sense, some of the results presented in this paper, such as those in section 5.1, are not surprising. We hope that you would agree with our view that presenting these results is scientifically important, even if they are somewhat negative and unsurprising.
>
> > **Q3:** OOD generalization.
>
> We evaluated our data on challenging and OOD settings in both real and simulated data. These videos include scenarios such as objects sliding down ramps, falling into containers, or domino-like setups. A summary of our relative improvement over the base OpenSora model is shown below.
>
> | Scenarios             | L2     | CD     | IoU     |
> | --------------------- | ------ | ------ | ------- |
> | Domino (Real)         | 47% | 54% | 42%  |
> | Ramp (Real)           | 41% | 47% | 18%  |
> | Stairs (Real)         | 25% | 20% | 103% |
> | Ramp (Simulated)      | 87% | 90% | 55%  |
> | Container (Simulated) | 75% | 67% | 3.7%   |
>
> Please see [here](https://anonymous.4open.science/r/ICML-7650-Rebuttal-1645/3_ood.md) for more information about the dataset construction and a full table breakdown across settings and comparisons with baselines.
>
> > **Q4:** Degradation of original capabilities.
>
> We evaluated our model on VBench to understand the effect that our post-training process has on the model's original capabilities. Overall, there is a degradation in aesthetic quality and image quality, likely stemming from the limited realism of our simulated videos. The degradations could potentially be mitigated by adding aesthetic samples into post-training, which has been shown to be effective for both image [2] and video [3] models. Aside from these degradations, the performance on the other metric categories either improves or remains intact. Please see [here](https://anonymous.4open.science/r/ICML-7650-Rebuttal-1645/6_vbench.md) for a full breakdown.
>
> > **Q5**: Concern about resolution.
>
> Due to computational limitations, we were not able to train models at resolution higher than 256. However, the video architecture of OpenSora supports zero shot evaluation at 512 resolution. Overall, our metrics slightly degrade, though performance could be improved with finetuning at 512. A summary on real data is shown below and the full table can be found [here](https://anonymous.4open.science/r/ICML-7650-Rebuttal-1645/5_512_resolution.md).
>
> | Output Resolution | L2 (⬇️) | CD (⬇️) | IoU (⬆️) |
> | ----------------- | ----------------- | ----------------- | ---------------- |
> | $256\times256$    | 0.153             | 0.432             | 0.069            |
> | $512\times512$    | 0.175             | 0.502             | 0.069            |
>
>
> Please let us know if you have any further questions. If our response has sufficiently addressed the concerns you have mentioned, we kindly ask that you raise your score. Thank you very much again for taking the time to review our work!
>
> References
>
> [1] Embers of autoregression show how large language models are shaped by the problem they are trained to solve
>
> [2] Emu: Enhancing Image Generation Models Using Photogenic Needles in a Haystack
>
> [3] Stable Video Diffusion: Scaling Latent Video Diffusion Models to Large Datasets

---

### Official Review · Reviewer_7wqN · 2025-03-18

**Overall Recommendation:** 2

**Summary:**

This work finds the generations of SOTA video generation models are visually impressive but are physically inaccurate. This work rigorously examines the post-training process of video generation models by focusing on the simple yet fundamental physics task of modeling object freefall which is highly challenging for state-of-the-art models.

They find fine-tuning a small amount of simulated videos can effectively improve the physical consistency. They further introduce two reward models for reward gradient training. The study also reveals key limitations of post-training in generalization and distribution modeling. Their released benchmark can also serve as a useful diagnostic tool for measuring the emergence of accurate physics modeling in video generative models.

Overall, the proposed benchmark is quite valuable for evaluating physics modeling, but the technical contribution is lacking and claims are not quite solid.

**Claims And Evidence:**

The claims are good overall.

A few issues:
- The improvement from ORO is quite marginal. For example, ORO increases IoU scores from 0.139 to 0.142. It's uncertain if the proposed reward models are useful or not.
- The authors conclude by experimenting on a single model Open-Sora, which is quite weak nowadays. It's unknown if the observations can transfer to other models.
- The paper doesn't evaluate if the learned physics can transferred to more OOD settings, i.e. dramatically different scenes (e.g. indoor, oudoor) and objects (e.g. cat, human etc.).

**Essential References Not Discussed:**

N/A

**Experimental Designs Or Analyses:**

Overall, the experimental design and analyses are solid and convincing.

A few issues:
- When ablation on dataset size, all models are trained for 5k steps, which may be under-trained for large dataset size.
- The fine-tuning video resolution is unspecified in the paper. From supplementary material, the models are trained on 256p, which may be too small for many objects in the proposed benchmark. How does the model perform on higher resolution, e.g. 512p?

**Methods And Evaluation Criteria:**

-  The proposed methods (PSFT, ORO) make sense and can improve the physical consistency.
- While the proposed reward models make sense, it's quite specialized and may not work for general cases. For example, does these reward models work for human actions?

**Other Comments Or Suggestions:**

N/A

**Other Strengths And Weaknesses:**

N/A

**Questions For Authors:**

See issues in previous sections.

**Relation To Broader Scientific Literature:**

N/A

**Theoretical Claims:**

No theoretical claims.

---

> ### Author Rebuttal · Authors · 2025-04-01
>
> We thank the reviewer for taking the time to review our work. We are extremely grateful for the feedback given, and below we address the main concerns raised.
>
> > **Q1:** Marginal improvement from ORO.
>
> On simulated data, ORO yields substantial gains. As the goal of our work is to study the process of post-training in a rigorous and controlled manner, we chose simulation as the primary domain for both training and evaluation. Even though we do not directly address the sim2real gap in this work, our method does also show generalizability to real data, and importantly, it maintains its outperformance of all other state-of-the-art commercial models.
>
> To see if we could push real world performance even further, we ran an experiment using depth reward. The depth reward improves performance on the real world dataset by a large margin in L2 (11% improvement) and Chamfer distance (15% improvement). Please see [here](https://anonymous.4open.science/r/ICML-7650-Rebuttal-1645/1_depth_reward.md) for the full results.
>
> We also found that ORO significantly improves the results in the phase of trajectory *after* collision with the ground, which is the most complex part to model. Please see our response to hCaK for more details.
>
> > **Q2:** Weakness of OpenSora.
>
> We agree that it is important to make sure the claims made for OpenSora hold for other models. We applied the PSFT procedure to Pyramid-Flow[1], a more recent and performant open video model than OpenSora. On our real test set, Pyramid-Flow outperforms the baselines.
>
> | Model               | L2 ⬇️ | CD ⬇️  | IoU ⬆️ |
> | ------------------- | ----------------- | ----------------- | ---------------- |
> | Pyramid-Flow + PSFT | 0.081         | 0.194         | 0.121        |
>
> Training curves and more evaluation tables can be found [here](https://anonymous.4open.science/r/ICML-7650-Rebuttal-1645/2_pyramid_flow.md).
>
> > **Q3:** OOD evaluation.
>
> We evaluated our data on challenging and OOD settings in both real and simulated data. These videos include scenarios such as objects sliding down ramps, falling into containers, or domino-like setups. A summary of our relative improvement over the base OpenSora model is shown below.
>
> | Scenarios             | L2     | CD     | IoU     |
> | --------------------- | ------ | ------ | ------- |
> | Domino (Real)         | 47% | 54% | 42%  |
> | Ramp (Real)           | 41% | 47% | 18%  |
> | Stairs (Real)         | 25% | 20% | 103% |
> | Ramp (Simulated)      | 87% | 90% | 55%  |
> | Container (Simulated) | 75% | 67% | 3.7%   |
>
> Please see [here](https://anonymous.4open.science/r/ICML-7650-Rebuttal-1645/3_ood.md) for more information about the dataset construction and a full table breakdown across settings and comparisons with baselines.
>
> > **Q4:** Reward method is specialized.
>
> We disagree that our reward modeling framework is specialized. In fact, it is highly general, since it only requires dense annotation maps, such as segmentation, depth or flow, to be provided for the generated and ground truth video.
>
> Accuracy on physics tasks besides dropping is an important problem though, and concurrent work has shown evidence that ORO could be effective in more general settings. VideoJam [2] uses optical flow supervision in a similar manner to ORO and finds that it dramatically improves motion accuracy in video diffusion models, including in modeling human motion.
>
> > **Q5:** Potential for under-training on larger datasets.
>
> We continued to finetune our model trained on 20k data samples for 5k more steps (batch size of 128). As can be seen in the figure [here](https://anonymous.4open.science/r/ICML-7650-Rebuttal-1645/4_20k_curve.md), our metrics do not consistently improve further as a result of doing this.
>
> > **Q6:** Concern about resolution.
>
> Due to computational limitations, we were not able to train models at resolution higher than 256. However, the video architecture of OpenSora supports zero shot evaluation at 512 resolution. Overall, our metrics slightly degrade, though performance could be improved with finetuning at 512. A summary on real data is shown below and the full table can be found [here](https://anonymous.4open.science/r/ICML-7650-Rebuttal-1645/5_512_resolution.md).
>
> | Output Resolution | L2 (⬇️) | CD (⬇️) | IoU (⬆️) |
> | ----------------- | ----------------- | ----------------- | ---------------- |
> | 256x256    | 0.153             | 0.432             | 0.069            |
> | 512x512    | 0.175             | 0.502             | 0.069            |
>
>
> Please let us know if you have any further questions. If our response has sufficiently addressed the concerns you have mentioned, we kindly ask that you raise your score. Thank you very much again for taking the time to review our work!
>
> References
>
> [1] Pyramidal Flow Matching for Efficient Video Generative Modeling
>
> [2] VideoJAM: Joint Appearance-Motion Representations for Enhanced Motion Generation in Video Models

---

### Decision · Program_Chairs · 2025-05-01

**Decision:**

Accept (poster)

**Comment:**

1x weak reject, 2x weak accept, 1x accept.
This submission introduces a diagnostic benchmark for assessing physical consistency in video generative models by focusing on object freefall, and it proposes post-training techniques—Physics-Supervised Fine-Tuning (PSFT) and Object Reward Optimization (ORO)—to improve physical realism.
The reviewers agree on the (1) value of the novel benchmark for revealing the physical limitations in current video generation models, (2) effectiveness of post‑training in mitigating major physical inaccuracies, and (3) comprehensive experimental design covering real and simulated data, supported further by additional human evaluation in the rebuttal.
They also concur on the (1) narrow scope limited solely to object freefall, (2) marginal improvements in some metrics (e.g. minor IoU gains), and (3) unresolved concerns regarding broad generalization and potential degradation of the base model’s overall performance.
Although Reviewer 7wqN remains less convinced despite the detailed author rebuttal, the strong follow-up responses provided by the authors—especially addressing limited scope, evaluation in diverse scenarios, and training detail clarifications—in combination with the more positive stances from the other reviewers, lead the AC to lean towards acceptance of this submission.